# The pluripotency factor NANOG controls primitive hematopoiesis and directly regulates *Tal1*

Julio Sainz de Aja[1], Sergio Menchero[1], Isabel Rollan[1], Antonio Barral[1], Maria Tiana[1], Wajid Jawaid[2,3], Itziar Cossio[1], Alba Alvarez[1], Gonzalo Carreño-Tarragona[1,4], Claudio Badia-Careaga[1], Jennifer Nichols[2,5], Berthold Göttgens[2,3] (ID), Joan Isern[1,6] (ID) & Miguel Manzanares[1,*] (ID)

## Abstract

Progenitors of the first hematopoietic cells in the mouse arise in the early embryo from *Brachyury*-positive multipotent cells in the posterior-proximal region of the epiblast, but the mechanisms that specify primitive blood cells are still largely unknown. Pluripotency factors maintain uncommitted cells of the blastocyst and embryonic stem cells in the pluripotent state. However, little is known about the role played by these factors during later development, despite being expressed in the postimplantation epiblast. Using a dual transgene system for controlled expression at postimplantation stages, we found that *Nanog* blocks primitive hematopoiesis in the gastrulating embryo, resulting in a loss of red blood cells and downregulation of erythropoietic genes. Accordingly, *Nanog*-deficient embryonic stem cells are prone to erythropoietic differentiation. Moreover, *Nanog* expression in adults prevents the maturation of erythroid cells. By analysis of previous data for NANOG binding during stem cell differentiation and CRISPR/Cas9 genome editing, we found that *Tal1* is a direct NANOG target. Our results show that *Nanog* regulates primitive hematopoiesis by directly repressing critical erythroid lineage specifiers.

**Keywords** erythropoiesis; gastrulation; *Nanog*; primitive hematopoiesis; *Tal1*
**Subject Categories** Development & Differentiation; Transcription
**The EMBO Journal (2019) 38: e99122**

## Introduction

Blood cells first appear during mouse development in the extraembryonic yolk sac at embryonic day (E) 7.5. These are primarily erythroid cells, needed to provide oxygen for the exponential embryo growth at these stages (Baron *et al*, 2012). This initial primitive hematopoiesis is not generated by hematopoietic stem cells, which first appear later in development (around E10.5) and provide the basis for definitive hematopoiesis (Jagannathan-Bogdan & Zon, 2013).

The precursors of the first erythroid cells are already present at the initial stages of gastrulation, in the nascent mesoderm at the posterior end of the embryo (Lawson *et al*, 1991; Huber *et al*, 2004); moreover, detailed fate mapping suggests that these cells are specified in the epiblast before gastrulation (Kinder *et al*, 1999; Padron-Barthe *et al*, 2014). Hematopoietic precursors are specified after the determination of the early mesoderm from the epiblast, which is driven by the sequential action of the transcription factors encoded by *Brachyury* and *Mesp1* and ends in the expression of FLK1 (encoded by *Kdr*), which marks most mesodermal cells at gastrulation (Pfister *et al*, 2007; Chan *et al*, 2013; Scialdone *et al*, 2016). Subsequently, primitive hematopoiesis progenitors start expressing a battery of lineage-specific transcription factor genes such as *Tal1*, *Gata1*, and *Klf1* as they migrate to the extraembryonic region and generate the blood islands of the yolk sac (Dore & Crispino, 2011; Baron *et al*, 2012).

Despite the knowledge acquired in recent years on the regulation of gastrulation and lineage determination of blood cells, we still do not fully understand how hematopoietic precursors are specified from within the pool of common mesodermal cells present in the posterior-proximal region of the gastrulating embryo. In other words, it remains unclear how the first differentiated cell type to appear in the postimplantation embryo (the primitive hematopoietic cells) is specified from a multipotent population of mesodermal progenitors, and how lineage-specific genes are turned on in this rapid transition. In this study, we provide evidence for an involvement in this process of the homeobox transcription factor gene *Nanog*.

NANOG is a constituent of the core set of transcription factors, together with OCT4 and SOX2, involved in establishing and maintaining embryonic pluripotency, both in the blastocyst and in embryonic stem (ES) cells in culture (Chambers & Tomlinson, 2009). Loss of *Nanog* in the early blastocyst results in embryonic

1   Centro Nacional de Investigaciones Cardiovasculares Carlos III (CNIC), Madrid, Spain
2   Wellcome-Medical Research Council Cambridge Stem Cell Institute, Cambridge, UK
3   Department of Haematology, Cambridge Institute for Medical Research, University of Cambridge, Cambridge, UK
4   Department of Haematology, Hospital 12 de Octubre, Madrid, Spain
5   Department of Physiology, Development and Neuroscience, University of Cambridge, Cambridge, UK
6   Department of Experimental & Health Sciences, University Pompeu Fabra (UPF), Barcelona, Spain
    *Corresponding author. Tel: +34914531200; E-mail: mmanzanares@cnic.es

death at implantation (Mitsui *et al*, 2003); however, *Nanog*-deficient ES cells are still able to maintain pluripotency, although they are prone to differentiate (Chambers *et al*, 2007). In the preimplantation embryo, *Nanog* is expressed throughout the epiblast. During implantation, *Nanog* is turned off, only to be re-expressed at E6.0 in the posterior part of the epiblast, where the primitive streak will form and gastrulation takes place shortly after (Hart *et al*, 2004; Osorno *et al*, 2012). Later, expression is restricted to primordial germ cells, with *Nanog* playing a crucial role in their development (Chambers *et al*, 2007; Yamaguchi *et al*, 2009; Zhang *et al*, 2018). Aside from its function in the germline, there is little or no previous evidence for *Nanog* playing any other role in the postimplantation epiblast or in the gastrulating embryo.

Here, we show that sustained expression of *Nanog* beyond gastrulation blocks differentiation of red blood cells during primitive hematopoiesis. This phenotype can be recapitulated in the adult, where *Nanog* leads to an increase in the number of megakaryocyte–erythroid precursors (MEPs), possibly by blocking their differentiation. Hematopoietic differentiation of *Nanog*-deficient ES cells is enhanced, further supporting the hypothesis that *Nanog* blocks the erythroid lineage in the epiblast of the gastrulating embryo. Furthermore, by re-analyzing single-cell RNA-seq data from gastrulating embryos (Scialdone *et al*, 2016) and NANOG ChIP-seq data in ES and epiblast-like cells (Murakami *et al*, 2016), together with CRISPR/Cas9-mediated genome editing, we found that NANOG directly represses the expression of the erythroid specifier *Tal1*. Together, these findings suggest that *Nanog* controls the early specification of hematopoietic cells from mesodermal precursors during gastrulation.

## Results

### *Nanog* blocks erythropoiesis in developing mouse embryos

*Nanog* loss of function is lethal at preimplantation stages (Mitsui *et al*, 2003), therefore preventing analysis of the putative role of Nanog later in development, when it is re-expressed at the posterior part of the gastrulating embryo (Hart *et al*, 2004). To overcome this obstacle, we used an inducible TetON transgenic model (*Nanog^tg*) in which *Nanog* expression is induced by the administration of doxycycline (dox) (Piazzolla *et al*, 2014). We induced *Nanog* from E6.5 in order to prolong its expression beyond E7.5, when it is normally turned off (Hart *et al*, 2004), and examined the embryos at E9.5. Visual examination of freshly dissected dox-treated embryos showed some growth retardation and craniofacial defects, but the most notable effect was a lack of blood (Fig 1A). To confirm this observation, we carried out whole-mount *in situ* hybridization for *Hbb-bh1*, which encodes the beta-like embryonic hemoglobin (Wilkinson *et al*, 1987) and for *Redrum*, an erythroid-specific long non-coding RNA (Alvarez-Dominguez *et al*, 2014; Paralkar *et al*, 2014). In untreated (control) *Nanog^tg* embryos at E9.5, *Hbb-bh1* labels primitive red blood cells that are distributed throughout the yolk sac. Expression of *Nanog* up to this stage resulted in near complete blockade of *Hbb-bh1* expression (Fig 1A). *Redrum* is expressed in the developing aorta-gonad-mesonephros (AGM) region, surely from erythroid cells circulating along the aorta, and in the tail bud. *Nanog* induction led to loss of *Redrum* expression in the AGM region, but interestingly not in the tail bud that is not a site of

embryonic erythropoiesis (Fig 1A). We also checked if the apparent lack of blood was accompanied by vascular defects. Immunostaining for Endomucin, expressed in embryonic endothelial cells, revealed no substantial differences at E9.5 between dox-treated and untreated *Nanog^tg* embryos, as is observed in the correct patterning of intersomitic vessels (Fig 1B). Furthermore, CD31 staining showed that yolk sac vasculature was equally unaffected in dox-treated embryos (Fig EV1A). We also examined heart morphology at these stages, to address if other mesodermal derivatives showed developmental defects. Hearts of freshly dissected E9.5 dox-treated embryos beat normally, and both overall morphology and histological sections showed no defects (Fig EV1B). Prolonged *Nanog* expression in the embryo thus causes a deficit in primitive red blood cells that is accompanied by lack of expression of erythroid-specific genes, but does not affect early vascular or cardiac development.

To characterize the effect of *Nanog* induction on hematopoiesis, we analyzed progenitors and red blood cells by flow cytometry of dispersed individual yolk sacs from E9.5 embryos using c-Kit (a marker of early uncommitted progenitors), CD41 (erythroid progenitors; Mitjavila-Garcia *et al*, 2002), CD71, and Ter119 (Borges *et al*, 2012). Dox-treated *Nanog^tg* embryos showed a dramatic reduction in erythroblast cells (CD71$^+$ Ter119$^+$; Fig 1C and D), which supports the above results. However, the total number of hematopoietic progenitor populations (cKit$^+$CD41$^+$ and CD41$^+$, respectively) remained unchanged (Fig 1E and F). We examined the morphology of erythroblasts from circulating blood of E9.5 dox-treated and untreated embryos by Giemsa staining (Fraser *et al*, 2007) and found that the few remaining primitive erythroid cells showed a normal morphology (Fig EV1C). Taken together, these results suggest that *Nanog* causes a blockade in hematopoietic progenitors, preventing their differentiation toward erythroblast cells.

### *Nanog* downregulates the expression of key erythroid determination genes

We next investigated how prolonged *Nanog* expression to E9.5 influences hematopoietic gene expression. For this, we isolated progenitor and mature populations by flow cytometry as described above (Fig 1C and D), and conducted RT–qPCR to examine the expression of core lineage determinants of hematopoietic fate: *Tal1*, *Runx1*, *Gata1*, and *Klf1* (Palis *et al*, 1999; Yokomizo *et al*, 2008; Kuvardina *et al*, 2015). Gain of *Nanog* expression in erythroblasts (CD71$^+$ Ter119$^+$ population) resulted in significant downregulation of *Tal1* and increase of *Runx1* (Fig 1G). However, despite consistent gain of *Nanog* expression, we did not observe differences of gene expression in earlier progenitors (Fig EV1D).

To examine whether similar changes occur at earlier stages, we induced *Nanog* expression from E5.5 to E7.5, a time window spanning initiation of primitive hematopoiesis. Whole-mount *in situ* hybridization showed decreased expression of *Gata1* and *Klf1* in the extraembryonic region, corresponding to the blood island domain (Fig EV1E). RT–qPCR of individual dox-treated or control E7.5 *Nanog^tg* embryos showed decreased expression of the core erythropoietic genes *Tal1*, *Gata1*, and *Klf1* but no change in *Runx1* (Fig EV1F). A possible explanation for our observations would be that *Nanog* expression causes a general blockade of mesodermal specification, with the downregulation of early hematopoiesis genes being merely a secondary effect of this. We therefore tested the

expression of lineage determinants expressed at gastrulation (*Brachyury* and *Eomes*) and the early mesodermal gene *Kdr* (Shalaby *et al*, 1995; Palis *et al*, 1999). Exogenous *Nanog* induced the expression of both *Brachyury* and *Eomes*, in line with published data (Teo *et al*, 2011), but did not alter *Kdr* expression (Fig EV1F). Together, these results suggest that *Nanog* blocks erythroid fate and is able to specifically downregulate the early expression of erythropoietic genes during the initial determination of primitive hematopoiesis.

### *Nanog*-induced hematopoietic defects are cell intrinsic

The results presented so far suggest that *Nanog* blocks specifically erythroid progenitors during primitive hematopoiesis. To test if this is the case, we aimed to rescue the observed genotype by generating chimeric embryos by injection of wild-type ES cells constitutively expressing GFP (Diaz-Diaz *et al*, 2017) into *Nanog*$^{tg}$ blastocysts. The resulting embryos were treated *in utero* with dox at E6.5 and examined for GFP fluorescence at E10.5. Those showing no overall contribution (no GFP$^+$ cells) were used as controls, whereas embryos containing GFP$^+$ cells were considered chimeras (Fig 2A and B). Erythroid cells were evaluated in individual embryos by flow cytometry analysis of CD71 and Ter119, as described earlier (Fig 1E and F).

Chimeras with high contribution of wild-type ES cells had circulating blood in both the embryo and the yolk sac, despite dox treatment, contrasting with embryos with no contribution (Fig 2B). Chimeras showed a recovery of erythroid cells, with high contribution from GFP$^+$ wild-type ES-derived cells (Fig 2C). Quantification of erythroid populations in chimeras showed an increased content of CD71$^+$ Ter119$^+$ cells (Fig 2D); this increase did not occur when the experiment was repeated without dox treatment (Fig 2E). The number of GFP$^-$ cells (derived from *Nanog* expressing cells) in dox-treated chimeras did not differ from that in controls (with no contribution of GFP$^+$ cells), demonstrating that the recovery of the erythroid populations in chimeras was entirely due to the wild-type ES cells (Fig 2F). These results indicate that the effect of *Nanog* on erythroid progenitors is primarily cell autonomous, and not secondary to *Nanog*-induced changes in other cell types.

### Loss of *Nanog* enhances hematopoietic differentiation of ES cells

To investigate the effect of the absence of *Nanog* on the erythroid lineage, we tested the potential of ES cells with homozygous *Nanog* loss of function (Chambers *et al*, 2007) to differentiate into blood cells in culture (Irion *et al*, 2010). *Nanog*$^{-/-}$ and wild-type control ES cells of the parental strain (E14Tg2a) were used to generate embryoid bodies (EB). EBs were allowed to differentiate for up to 7 days in hematopoietic differentiation media. After disaggregation and culture, different colony-forming units (CFU) were scored between days 5 and 7 (D5–D7; Fig 3A). Despite a trend for a decrease in the number of common myeloid progenitors (CFU-GEMM), *Nanog*$^{-/-}$ EBs generated significantly more primitive erythroid colonies (Ery-P) than controls, as well as a significantly higher number of mature erythroid colonies (BFU-E; burst forming unit erythroid) in the presence of cytokines driving a broader hematopoietic differentiation. Interestingly, there was no between-genotype difference in granulocyte–monocyte (CFU-GM) progenitors, but monocyte (CFU-M) or granulocyte (CFU-G) progenitors were produced more abundantly from wild type than from *Nanog*$^{-/-}$ EBs (Fig 3A). This last observation is possibly due to a decrease in common myeloid progenitors together with a significant increase of erythroid progenitors in the mutants. *Nanog*$^{-/-}$ ES cells thus have an increased potential for specific differentiation to red blood cells.

To investigate how the absence of *Nanog* affects the gene networks involved in erythroid specification, we monitored control and *Nanog*$^{-/-}$ ES-derived EBs for the expression of selected markers over 10 days of differentiation. *Brachyury* expression was examined as a marker of initial mesoderm specification, a necessary first step for the establishment of hematopoietic lineages in EBs (Fehling *et al*, 2003). *Brachyury* expression markedly increased at day 3 in wild-type cells, as previously described (Robertson *et al*, 2000), but in *Nanog*$^{-/-}$ EBs this expression peak was delayed until day 5 (Fig 3B). *Nanog* is thus likely required for the correct temporal activation of *Brachyury*. We next checked the expression of genes encoding the erythroid-specific factors *Tal1*, *Gata1*, and *Klf1* and the embryonic hemoglobin gene *Hbb-bh1*. In wild-type EBs, erythroid gene expression peaks around day 5, 2 days after *Brachyury* activation. In *Nanog*$^{-/-}$ EBs, erythroid gene expression peaked a day later, at day 6. However, this is only 1 day after the onset of *Brachyury* expression, contrasting the 2-day delay in wild-type EBs (Fig 3B). Given the requirement of *Brachyury* expression for hematopoietic differentiation (Fehling *et al*, 2003), we aligned the expression dynamics of wild-type and *Nanog*$^{-/-}$ cells to the day of *Brachyury* induction (Fig EV2A). To validate this approach, we

▶

---

**Figure 1. Effect of *Nanog* on erythropoietic development.**

A   Dox-induced prolongation of *Nanog* expression in *Nanog*$^{tg}$ embryos up to E9.5 results in lack of blood (left) and downregulation of erythropoietic gene expression. The center and right panels show whole-mount *in situ* hybridization for *Hbb-bh1* (in embryos with intact yolk sacs) and for the long non-coding RNA *Redrum*. Asterisks mark the aorta-gonad-mesonephros (AGM) region and arrows the tail bud. Embryos of the same genotype but not treated with dox were used as controls (−dox). Scale bars, 500 μm.

B   Endomucin staining of vessels in control (−dox) or treated (+dox) E9.5 *Nanog*$^{tg}$ embryos. On the right, higher magnifications of the boxed areas. Scale bar, 500 μm.

C   Representative FACS plot of the distribution of the CD71 and Ter119 populations in dissected yolk sacs from untreated and dox-treated E9.5 *Nanog*$^{tg}$ embryos.

D   Quantification of the CD71$^+$ Ter119$^+$ population in controls (−dox, black dots; $n = 8$) and *Nanog* expressing (+dox, red dots; $n = 7$) E9.5 yolk sacs. Each replicate contained a pool of 5 (−dox) or 8 (+dox) E9.5 *Nanog*$^{tg}$ embryos. \*\*\*$P < 0.0005$; Student's *t*-test. Horizontal line represents mean values and error bars standard deviation (SD).

E   Representative FACS plots showing the distribution of cKit and CD41 populations in yolk sacs from untreated controls (−dox) and *Nanog* expressing (+dox) E9.5 *Nanog*$^{tg}$ embryos.

F   Quantification of different progenitor populations in yolk sacs from control (−dox, black dots; $n = 8$) and *Nanog* expressing (+dox, red dots; $n = 7$) E9.5 embryos. Each replicate contained a pool of 5 (−dox) or 8 (+dox) E9.5 *Nanog*$^{tg}$ embryos. Horizontal line represents mean values and error bars SD.

G   Differences in the expression levels of *Nanog* and selected hematopoietic genes in the CD71$^+$ Ter119$^+$ population of control (−dox; $n = 7$) and *Nanog* expressing (+dox; $n = 4$) E9.5 embryos. \*\*$P < 0.005$, \*\*\*$P < 0.0005$; Student's *t*-test. Horizontal line represents mean values and error bars SD.

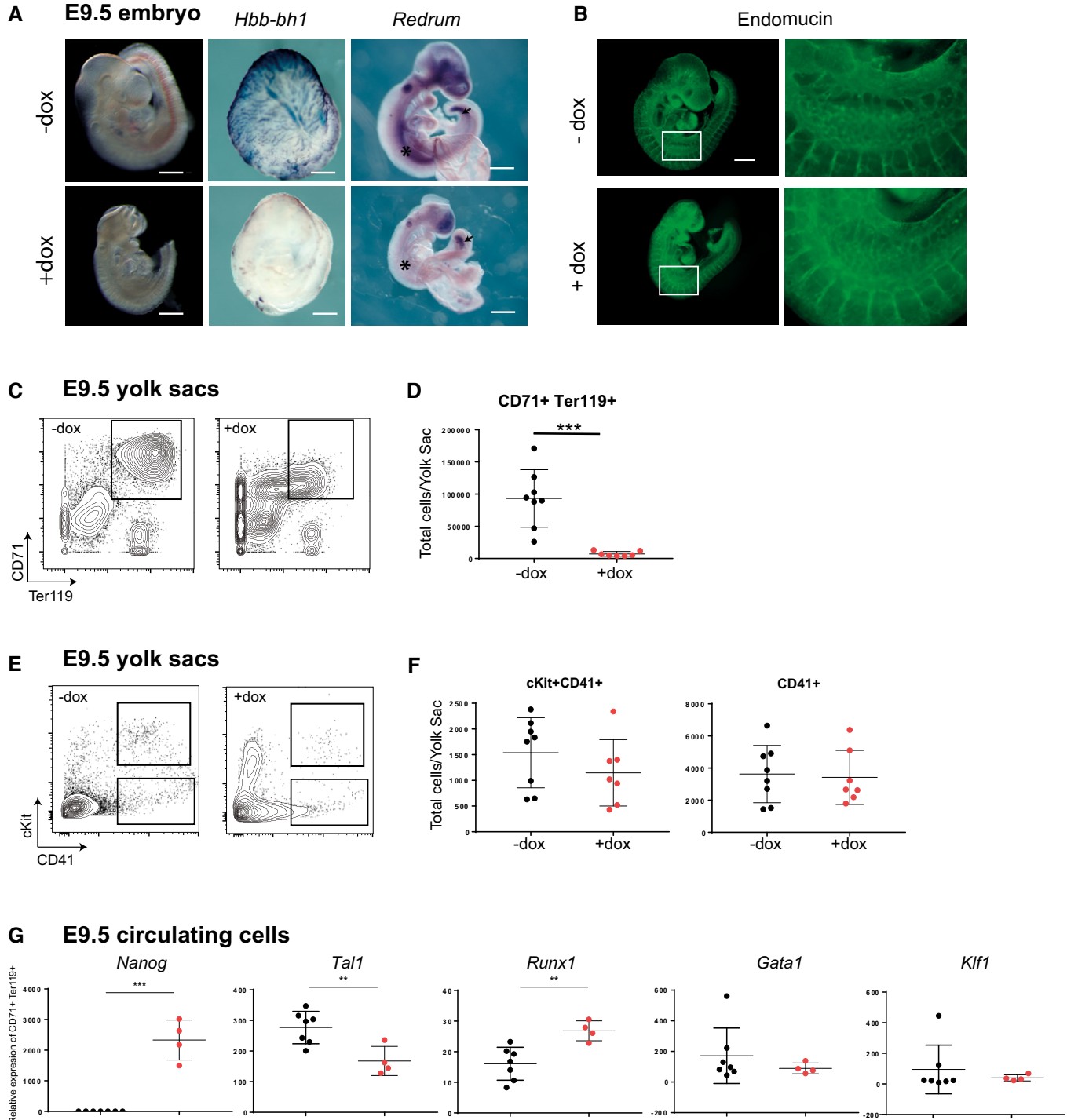

**Figure 1.**

examined the expression of *Kdr*, a pan-mesodermal gene that acts downstream of *Brachyury*; relative to the timing of *Brachyury* induction, dynamics of *Kdr* expression coincided in wild-type and *Nanog*$^{−/−}$ EBs. In contrast, erythroid gene activation occurred earlier in *Nanog*$^{−/−}$ EBs than in wild-type controls (Fig EV2B). Thus, although mesoderm induction is delayed in *Nanog*$^{−/−}$ EBs,

once it occurs the *Nanog*$^{−/−}$ mesodermal cells show an elevated potential for erythroid differentiation.

To further study the effect of loss of *Nanog*, we deleted a floxed allele from a heterozygous ES cell line (*Nanog*$^{flox/−}$; Zhang *et al*, 2018) by transfecting Cre recombinase and differentiating sorted GFP$^+$ cells (that is activated upon Cre recombination) from ES to

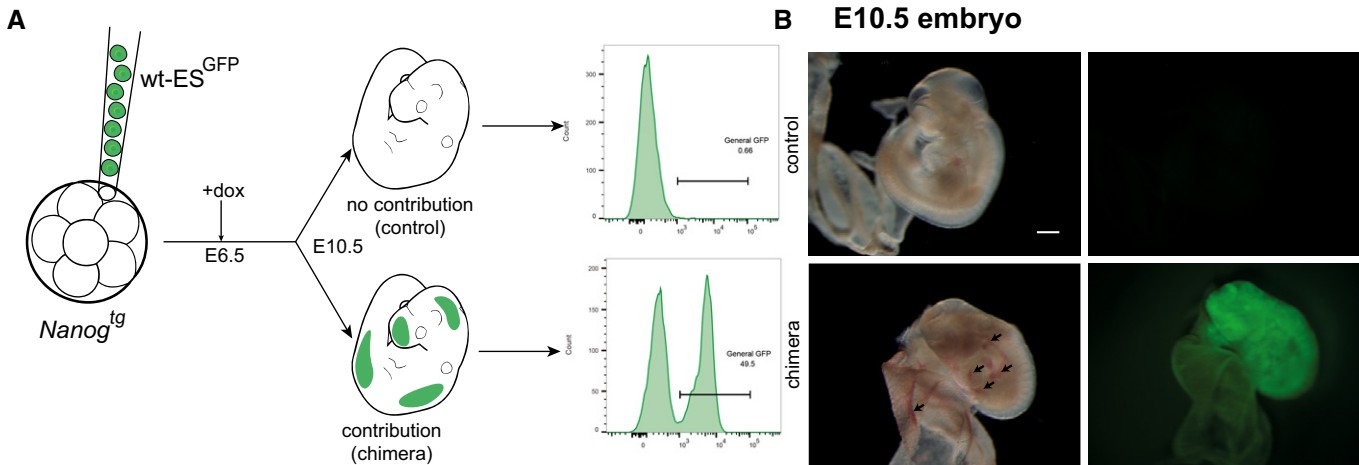

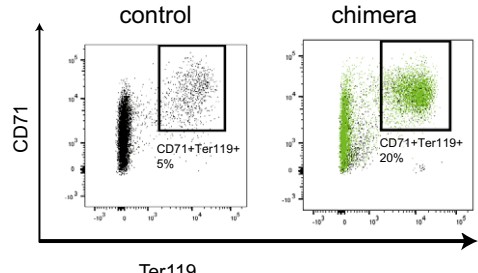

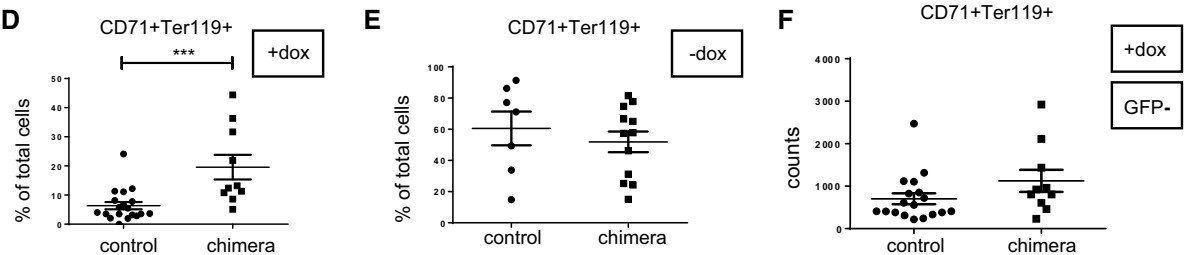

**Figure 2.  Wild-type ES cells rescue erythroid maturation in chimeric embryos.**

A     Experimental design for chimera generation, and contribution of GFP cells to chimeric embryo (right hand side panels).

B     Freshly dissected dox-treated *Nanog^tg* E10.5 embryos without (control) and with (chimera) contribution of wt-ES^GFP cells (left, brightfield; right, GFP). Arrows mark the presence of blood in chimeric embryos that is absent from controls. Scale bar, 500 μm.

C     Representative FACS plots showing of red blood cell maturation as determined by CD71/Ter119 staining in single dox-treated E10.5 control (left) and chimeric (right) embryos.

D–F     Quantification of the CD71+ Ter119+ population in single dox-treated E10.5 embryos (D; control, n = 18; chimera, n = 10), untreated embryos (E; control, n = 7; chimera, n = 12), and in GFP− cells (not derived from wild-type ES cells) from dox-treated embryos (F; control, n = 18; chimera, n = 10). ***P < 0.0005; Student's t-test. Horizontal line represents mean values and error bars SD.

epiblast-like cells (Hayashi *et al*, 2011; Murakami *et al*, 2016). This process recapitulates in culture the transition from pluripotent cells of the blastocyst to primed cells of the epiblast (Buecker *et al*, 2014), a time window during development when *Nanog* is expressed. Mutant cells (*Nanog^{del/−}*) upregulate Brachyury following the same dynamics as control heterozygote *Nanog^{flox/−}* cells. However, they show precocious activation of erythroid gene expression (Fig EV2C), in line with our previous observations.

### Blockade of adult erythrocyte maturation by *Nanog*

*Nanog* has mostly been analyzed in early developmental stages and in pluripotent stem cells. However, some reports have described its expression and roles in adult tissues and cells (Tanaka *et al*, 2007; Kohler *et al*, 2011; Piazzolla *et al*, 2014). In light of our findings during embryonic hematopoiesis, we explored the effects of *Nanog* during erythroid differentiation in the adult.

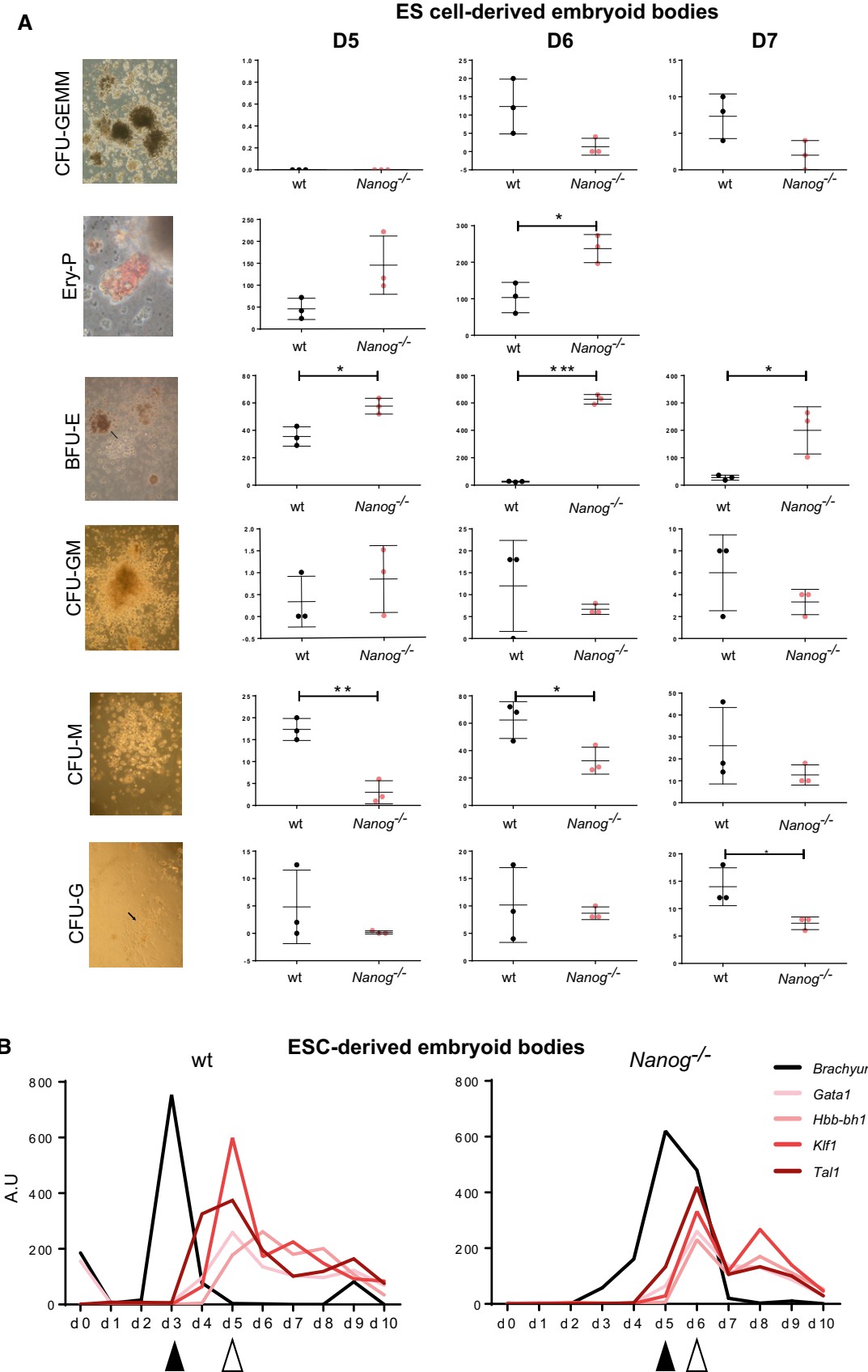

**Figure 3.**

**Figure 3.  *Nanog*-knockout ES cells show increased potential to generate erythroid precursors.**

A  Quantification of colony-forming units generated by wild-type (wt) and knockout ($Nanog^{-/-}$) ES cells after culture of EBs for 5 (D5), 6 (D6), or 7 (D7) days and plating disaggregated cells in different hemogenic-promoting conditions. Panels on the left show representative images of mouse hematopoietic colonies obtained after 12 days of culture in specific media. CFU-GEMM, progenitors giving rise to granulocytes, erythrocytes, monocytes, and megakaryocytes; BFU-E, burst forming units—erythroid; Ery-P, colony-forming primitive erythroid; CFU-GM, granulocyte–monocyte precursors; CFU-M, monocyte precursors; CFU-G, granulocyte precursors. No CFU-GEMM are detected at D5 and no BFU-E at D7. For both wt and knockout cells, $n = 3$ each with three technical replicates. \*$P < 0.05$, \*\*$P < 0.005$, \*\*\*$P < 0.00005$; Student's $t$-test. Horizontal line represents mean values and error bars SD.

B  RT–qPCR determination of the relative expression of *Brachyury* and selected hematopoietic genes in control (wt, right) and knockout ($Nanog^{-/-}$, left) ES cells ($n = 3$) during 10 days of EB differentiation in hematopoietic cytokine-enriched medium. Black arrowheads indicate the peak of *Brachyury* expression and white arrowheads the time of maximum hematopoietic gene expression.

*Nanog* expression was systemically induced in adult $Nanog^{tg}$ mice by 5-day treatment with dox in drinking water, and the mice were then sacrificed and bone marrow extracted (dox$^+$; Fig 4A). As controls, we used untreated mice of the same genotype (dox−). Analysis of erythrocyte maturation with CD71 and Ter119 (Socolovsky *et al*, 2001; Zhang *et al*, 2003) revealed an increase in immature populations (basophilic and polychromatic erythroblasts; S2 and S3, respectively) together with a decrease in the number of more differentiated erythroblasts (orthochromatic erythroblasts, S4; Fig 4B and C). This result suggested a block in the differentiation of erythrocyte precursors, so we next quantified bone marrow progenitors by flow cytometry using lineage cocktail, c-kit, Sca-1, CD34, and CD16/32 (Fig 4D; Challen *et al*, 2009).

Induced *Nanog* expression triggered a decrease in absolute cell numbers of hematopoietic stem cells (lineage-Sca1$^+$cKit$^+$; LSK) and common myeloid progenitors (CMP), but no changes in granulocyte–macrophage progenitors. Interestingly, this was accompanied by a significant increase in megakaryocyte–erythroid progenitors (MEP; Fig 4E). Analysis of the expression of key erythroid genes by RT–qPCR in sorted MEPs revealed a significant reduction of *Tal1* in dox-treated mice (Fig 4F). Together, these results indicate that *Nanog* can block the maturation of red blood cells during adult hematopoiesis together with the downregulation of key erythroid factors. This leads to defective differentiation of these populations and therefore to an accumulation of their precursors.

We further characterized this phenotype by RNA-seq on the MEPs from dox-treated and untreated adult $Nanog^{tg}$ mice. Genes downregulated in MEPs from dox-treated animals were enriched in functional terms related to bone marrow cell populations, and more specifically MEPs (Fig EV3A). This confirms that *Nanog* is repressing the transcriptional program for erythroid progenitors (Fig EV3B). On the other hand, genes that are upregulated upon *Nanog* induction are highly enriched in the mast cell program (Fig EV3A and B; Dataset EV1). Most interestingly, deletion of *Tal1* during adult hematopoiesis results in production of mast cells from MEPs, while under normal conditions these cells derive from granulocyte–monocyte progenitors (Salmon *et al*, 2007). This is accompanied by an upregulation of *Gata2* (Salmon *et al*, 2007), a critical specifier of mast cells (Ohmori *et al*, 2015), that we also see increased upon *Nanog* expression in MEPs (Fig EV3B; Dataset EV1). Furthermore, the expression of *Cebpa*, a factor that represses mast cell lineage (Iwasaki *et al*, 2006), is downregulated in the *Nanog*-expressing MEP population (Fig EV3B; Dataset EV1). However, we believe that the positive regulation of the mast cell program is not a physiological role of *Nanog*, because this cell type does not appear during gastrulation (as erythroid progenitors do) but at later stages in the yolk sac and the AGM (Gentek *et al*, 2018) where *Nanog* is

not expressed. Thus, we consider that upregulation of the mast cell program is a secondary consequence of the downregulation of erythroid lineage factors, such as *Tal1*, in *Nanog* expressing MEPs.

To extend these observations, we next carried out bone marrow transplantation of $Nanog^{tg}$ mice to wild-type irradiated recipients (Fig 4G). After 3 months of engraftment and recovery, more than 95% of peripheral blood cells were derived from donor mice ($n = 7$; Fig 4H). We treated the mice for 4 months with dox to induce *Nanog* expression only in hematopoietic cells, and found that at that point the host cells had been partially able to recolonize the bone marrow and contribute to peripheral blood cells (ranging from 20 to 80%; Fig 4H). We then purified bone marrow from the transplanted mice and analyzed chimerism in different progenitor populations. While LSK, CMPs of GMPs show variable degrees of contribution of wild-type cells and *Nanog* expressing cells, MEPs are almost exclusively derived from the host (Fig 4I). These results indicate that the expression of *Nanog* in MEPs causes them to be outcompeted by wild-type cells during bone marrow reconstitution, possibly due to their decreased ability to differentiate and generate mature erythroid cells.

### A distal NANOG-binding element represses *Tal1* expression in the embryo

*Nanog*-mediated downregulation of erythroid specification genes in both the embryo and the adult strongly suggests that some of these genes are likely direct transcriptional targets of NANOG. If so, we would expect to find mutually exclusive expression of *Nanog* and these genes at the time of initial hematopoietic specification in the gastrulating embryo. We therefore analyzed single-cell expression data from E7.0 nascent mesoderm (Scialdone *et al*, 2016), when *Nanog* is still expressed in the posterior-proximal region of the embryo (Hart *et al*, 2004), and examined the number of cells expressing both *Nanog* and markers of mesoderm (*Brachyury*, *Cdx2*) and hematopoiesis (*Tal1*, *Runx1*, *Gata1*, *Klf1*; Fig 5A). For all of these genes, we found the expected proportion of co-expressing cells with *Nanog* with the exception of *Tal1* (Fig 5A and B). We confirmed that *Nanog* can downregulate *Tal1* at early stages by culturing $Nanog^{tg}$ embryos with or without dox from E6.5 to E6.75 *ex utero*, which did not alter normal development (Fig 5C). *Tal1* failed to be upregulated in dox-treated embryos, whereas other hematopoietic genes such as *Klf1* were unaffected (Fig 5D). We further confirmed that *Nanog* downregulates *Tal1* by whole-mount *in situ* hybridization of E7.5 embryos treated with dox *in utero* (Fig 5E).

This evidence strongly suggests that *Tal1* is a direct transcriptional target of NANOG during early gastrulation at the onset of

hematopoietic determination. To investigate this possibility, we analyzed published ChIP-seq data for NANOG binding in ES and EpiLCs, which correspond to the E6.0 epiblast in the mouse embryo (Murakami *et al*, 2016). This study describes a broad resetting of

NANOG-occupied genomic regions in the transition from ES cells to EpiLCs, resembling the developmental progress from the naïve inner cell mass of the blastocyst to the primed epiblast at gastrulation (Hayashi *et al*, 2011; Morgani *et al*, 2017). We examined a number

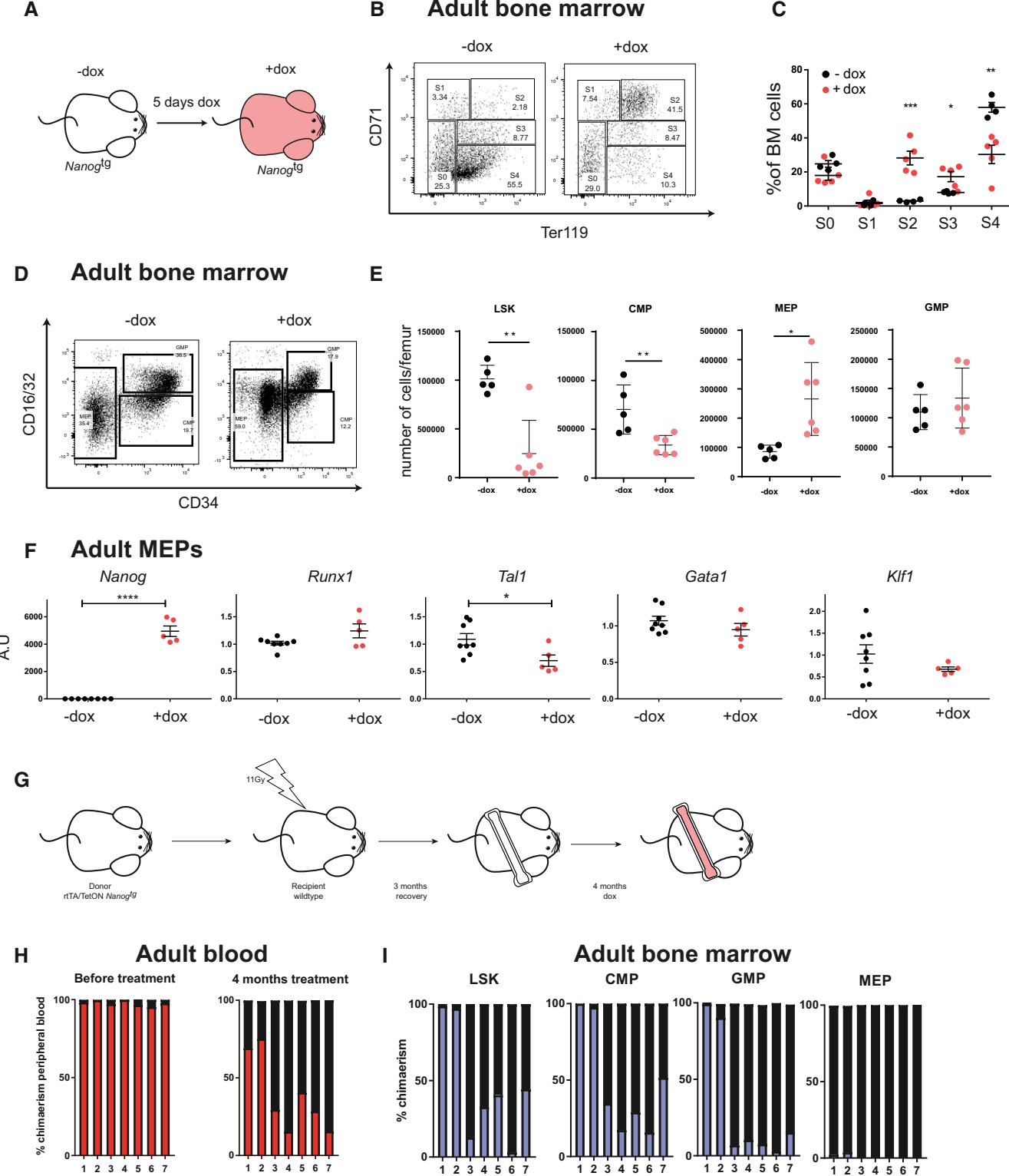

**Figure 4.**

**Figure 4.   Induced *Nanog* expression blocks erythroid maturation in adult mice.**

A   Experimental design for the treatment of adult *Nanog^tg^* mice.

B   Representative FACS plots showing the distribution of different populations distinguished by CD71/Ter119 staining in whole bone marrow from untreated (−dox) or treated (+dox) adult mice. S0 (double negative cell), S1 (proerythroblast), S2 (basophilic erythroblast), S3 (polychromatic erythroblast), and S4 (orthochromatic erythroblast) are different stages of blood maturation.

C   Quantification of the S1–S4 erythroid populations (−dox, $n = 4$; +dox, $n = 5$). *$P < 0.05$, **$P < 0.005$, ***$P < 0.0005$; Student's $t$-test. Horizontal line represents mean values and error bars SD.

D   Representative FACS plots showing the distribution of CD16/32 and CD34 hematopoietic precursors sorted from the cKit⁺Sca1⁻LIN⁻ bone marrow of untreated (−dox) or treated (+dox) adult *Nanog^tg^* mice.

E   Quantification of precursor populations based on CD16/32 and CD34 sorting, as total number of cells per individual femur (−dox, $n = 5$; +dox, $n = 6$). *$P < 0.05$, **$P < 0.005$; Student's $t$-test. Horizontal line represents mean values and error bars SD.

F   RT–qPCR quantification of the relative expression of hematopoietic genes in megakaryocyte–erythroid progenitors (MEP; −dox, $n = 8$; +dox, $n = 5$). *$P < 0.05$, ****$P < 0.00005$; Student's $t$-test. Horizontal line represents mean values and error bars SD.

G   Experimental design for the transplant of *Nanog^tg^* bone marrow to wild-type recipients and treatment of chimeric mice.

H   Contribution of *Nanog^tg^* transplanted bone marrow cells to peripheral blood before (left) and after (right) dox treatment. Percentage of host-derived cells (CD45.1⁺) are shown in black, and of donor derived cells (CD45.1/CD45.2 double +) in red. Individual mice are indicated on the $x$-axis ($n = 7$).

I   Contribution of *Nanog^tg^* transplanted cells to LSK, CMP, GMP, and MEP populations purified from bone marrow. Percentage of host-derived cells (CD45.1⁺) are show in black, and of donor derived cells (CD45.1/CD45.2 double +) in blue. Individual mice are indicated on the $x$-axis ($n = 7$).

of genomic loci, detecting binding at the *Nanog* locus itself in both ES cells and EpiLCs (Fig EV4A) and in *Cdx2* only in ES cells (Fig EV4B). Neither cell type showed evidence of NANOG bound regions surrounding *Runx1* (Fig EV4C) or *Klf1* (Fig EV4D). Interestingly, EpiLCs, but not ES cells, showed NANOG binding 22 kilobases upstream of *Tal1*, in an intron of the neighboring *Stil* gene (Fig EV4E). We also detected NANOG binding downstream of *Gata1* (Fig EV4F); however, these regions could be functionally related to the neighboring *Eras* and *Hdac6* genes, which are associated with pluripotency and early stem cell differentiation (Takahashi *et al*, 2003; Chen *et al*, 2013).

Analysis of the *Tal1/Stil* NANOG bound region in EpiLCs (Fig 5F) revealed bona fide consensus binding sites (Fig EV5A). To investigate the function of this region, we deleted it by CRISPR/Cas9-mediated genome editing (Ran *et al*, 2013) by microinjection in one-cell stage embryos and examined the transcriptional consequences in early development. Gene expression was analyzed by RT–qPCR in individual edited E6.5 embryos. As controls, we used embryos of the same batch showing no evidence of deletion of the *Tal1/Stil* NANOG bound region (Fig EV5B). *Tal1* expression was significantly increased in targeted embryos, whereas other genes such as *Klf1*, *Gfi1b*, or *Runx1* were unaffected (Fig 5G). Deletion of this genomic region did not alter *Stil* expression, despite the location of the site within this gene (Fig 5G). These assays provide strong evidence that this specific genomic region acts as a cis-regulatory element in the *Nanog*-mediated repression of *Tal1* in the early mouse embryo.

In order to further confirm these observations and address the effect of the deletion on *Tal1* expression and its dependence on *Nanog*, we analyzed the transition from ES to EpiLC in culture as above. For this, we generated lines deleted for the distal *Tal1* element by genome editing as previously described *in vivo*, but in ES cells derived from the *Nanog^tg^* mouse (Figs 5H and EV5C). We observe that non-treated *Nanog^tg^* ES cells (non-deleted control) show a significant increase in *Tal1* expression when they transit to EpiLCs (Fig 5H), what would be the equivalent of the initial expression of *Tal1* in the embryo. However, if dox is added to the medium, this increase of *Tal1* between ES and EpiLC is no longer significant. Thus, in this experimental setting, increased expression of *Nanog* is able to block at least partially the early induction of *Tal1*, in line

with our *in vivo* results. Nevertheless, when we repeat his experiment but with two independent ES cell lines where the NANOG-bound distal element (*dTal1*) has been deleted (*Nanog^tg^;dTal1^del#1^* and *Nanog^tg^;dTal1^del#2^*), EpiLC become unresponsive to *Nanog* upon dox treatment and still upregulate *Tal1* as cells not treated with dox. These results show that the distal element we have characterized is necessary for correct initiation of *Tal1* expression, and that it mediates the response of *Tal1* to *Nanog*.

## Discussion

Red blood cell precursors are the first cell type to be specified from nascent mesoderm during mouse gastrulation (Kinder *et al*, 1999; Baron *et al*, 2012). While the genes and networks that determine primitive hematopoietic cells are well understood (Isern *et al*, 2011; Kingsley *et al*, 2013), much less is known about how precursors are specified during the early stages of primitive streak formation (Padron-Barthe *et al*, 2014). Here, we show that the pluripotency factor NANOG regulates the transition from multipotent mesodermal progenitors to red blood cell precursors in these early steps, at least partially through the direct regulation of the lineage specifier *Tal1*.

Despite the well-characterized role of pluripotency factors in embryonic stem cells and the preimplantation embryo (Chambers & Tomlinson, 2009), their function at later developmental stages has received much less attention, even if they are expressed up to gastrulation in mice (Yeom *et al*, 1996; Hart *et al*, 2004; Osorno *et al*, 2012) and primates (Nakamura *et al*, 2016). *Oct4* is involved in proliferation of the primitive streak (DeVeale *et al*, 2013), in correct trunk elongation of the trunk (Aires *et al*, 2016), and some evidence points to it having a role in mesoderm and subsequent hematopoietic specification (Kong *et al*, 2009). However, no clear function is known for *Nanog* after implantation apart from the regulation of germline development (Chambers *et al*, 2007). Single-cell RNA-seq expression data from gastrulating embryos (Scialdone *et al*, 2016) show that *Nanog* is expressed in a subset of mesodermal precursors. This situation is reminiscent of the heterogeneities in *Nanog* expression in the preimplantation embryo, which drives lineage segregation of the epiblast and the primitive endoderm

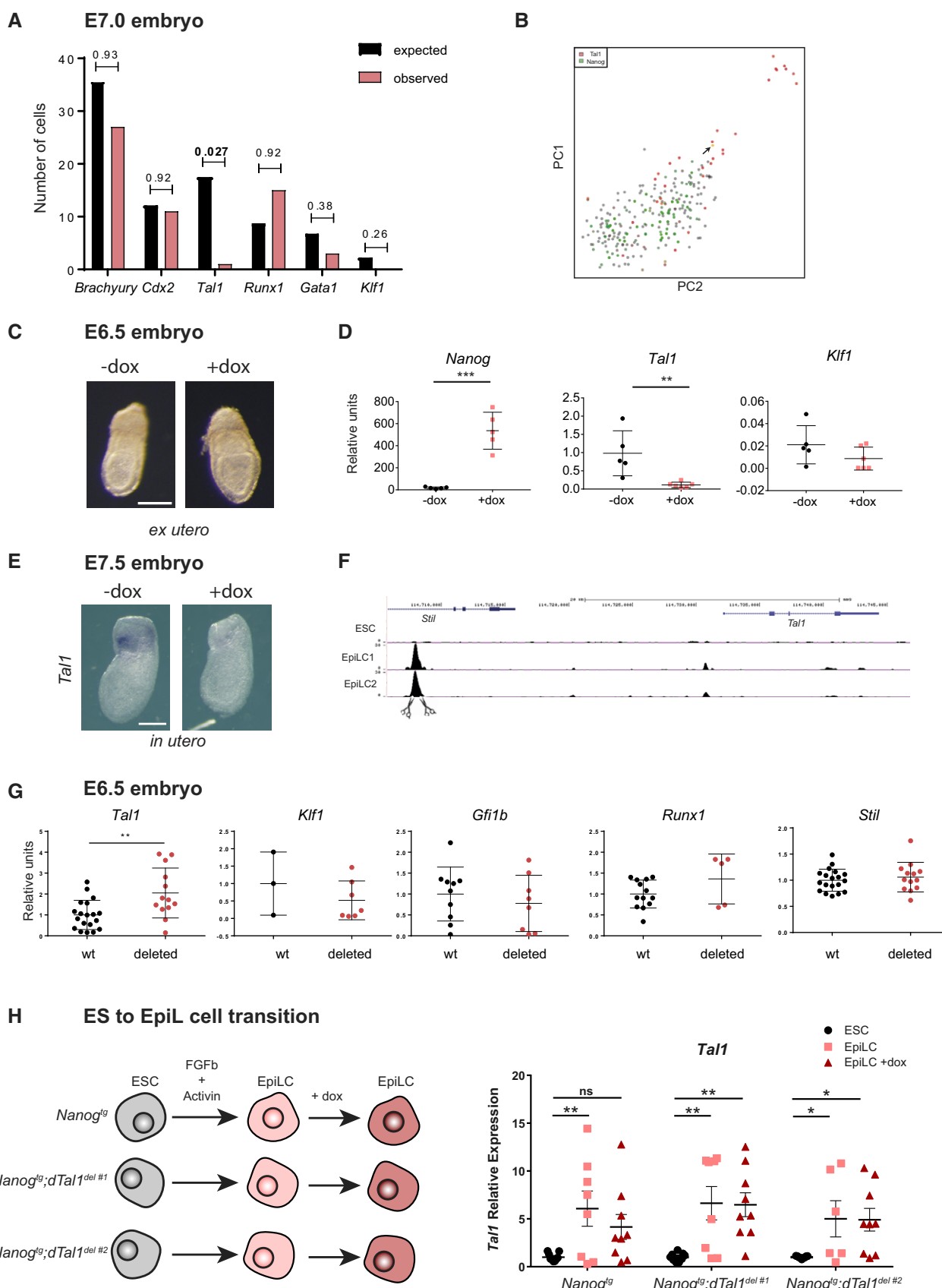

**Figure 5.**

**Figure 5.  Direct transcriptional regulation of *Tal1* expression by *Nanog*.**

A   Expected and observed number of mesodermal (Flk1⁺) cells of the E7.0 mouse embryo expressing *Nanog* and selected mesodermal or hematopoietic gene expression, based on single-cell RNA-seq data (Scialdone *et al*, 2016). Statistical significance was calculated with a hypergeometric test.

B   PCA showing the distribution of Flk1⁺ E7.0 mesoderm cells expressing *Nanog* (green) or *Tal1* (red). The single cell expressing both genes is shown in yellow and indicated by an arrow.

C   E6.5 *Nanog^tg* embryos after 8 h *ex utero* culture in the presence (+dox) or absence (−dox) of doxycycline. Scale bar, 100 μm.

D   RT–qPCR quantification of the relative expression of *Nanog*, *Tal1*, and *Klf1* in individual untreated embryos (−dox) or treated embryos (+dox) (*n* = 5). **$P < 0.005$, ***$P < 0.0005$; Student's *t*-test. Horizontal line represents mean values and error bars SD.

E   Whole-mount *in situ* hybridization of *Tal1* in E7.5 untreated (−dox) or *in utero* treated (+dox) *Nanog^tg* embryos. Scale bar, 100 μm.

F   UCSC browser view of the *Tal1/Stil1* region (mm9; chr4:114,705,753-114,756,741), indicating the presence of the NANOG binding peak, determined by ChIP-seq, in EpiLCs (2 replicates are shown) but not in ES cells (Murakami *et al*, 2016); the binding peak was deleted by CRISPR/Cas9 genome editing (scissors).

G   RT–qPCR determination of relative expression in wild-type and CRISPR-deleted embryos (*n* = 5) of *Tal1* (wt, *n* = 19; deleted, *n* = 13), *Klf1* (wt, *n* = 3; deleted, *n* = 6), *Gfi1b* (wt, *n* = 10; deleted, *n* = 8), *Runx1* (wt, *n* = 13; deleted, *n* = 5), and *Stil* (wt, *n* = 19; deleted, *n* = 13). **$P < 0.005$, Student's *t*-test. Horizontal line represents mean values and error bars SD.

H   Experimental design for ES to EpiL cell differentiation of *Nanog^tg* cells and two independent clones (*Nanog^tg;dTal1^del#1* and *Nanog^tg;dTal1^del#2*) where the binding site for NANOG distal to *Tal1* has been deleted (left). On the right, relative expression of *Tal1* determined by RT–qPCR for each ES cell line (ESC; *n* = 9 for all three lines) and EpiL cells without (EpiLC; *Nanog^tg* and *Nanog^tg;dTal1^del#1*, *n* = 8; *Nanog^tg;dTal1^del#2*, *n* = 6) or with dox treatment (EpiLC +dox; *n* = 9 for all three lines). The genotype of the cell lines is indicated below. Values were normalized to *Nanog^tg* ESC. *$P < 0.05$, **$P < 0.01$, ns = not significant; ANOVA with Fisher post-test. Horizontal line represents mean values and error bars standard error of the mean (SEM).

(Xenopoulos *et al*, 2015). Our results suggest that a similar situation may occur during specification of the first mesodermal lineages. *Nanog* expression in *Brachyury*-positive cells would maintain them in a pan-mesodermal multipotent state, whereas its downregulation would allow the expression of early hematopoietic lineage specifiers, driving their differentiation to primitive red blood cells. This process, however, occurs during a limited time window during the initial phases of gastrulation, as *Nanog* is quickly downregulated by E8.0–8.5 (Hart *et al*, 2004; Scialdone *et al*, 2016). By this stage, mesodermal progenitors have ingressed through the primitive streak and are no longer able to activate the early hematopoietic program, a process that also involves restricted spatial signaling through the Wnt and Bmp pathways (Cheng *et al*, 2008; Myers & Krieg, 2013; Mimoto *et al*, 2015). Therefore, this *Nanog*-mediated switch would act to control the rapid specification of blood precursors, the first lineage determination event in gastrulation, and required to supply the embryo with oxygen to support its subsequent exponential growth.

We also show that *Nanog* directly represses the master hematopoietic regulator *Tal1* (Porcher *et al*, 2017) through an upstream regulatory element located in an intron of the neighboring *Stil* gene. Interestingly, this site is occupied by NANOG only during the differentiation of ES cells to EpiLCs (Murakami *et al*, 2016). This change in binding site usage during this transition again suggests that *Nanog* has specific roles in the postimplantation pregastrulating epiblast (the *in vivo* equivalent of EpiLCs; Hayashi *et al*, 2011) that are distinct from those operating during the pluripotent state. *Tal1* is certainly a prime candidate for mediating at least partially the effects of *Nanog* on erythropoiesis, as we found that it is consistently repressed at different embryonic stages and in adult erythroid progenitors. However, surely other genes involved in early erythroid development, such as *Gata1*, could be also direct *Nanog* targets during this process. Further studies will unravel the full network regulated by *Nanog* at these stages.

In the adult, *Nanog* expression leads to defective erythroid-cell maturation, as also occurs in the embryo, and to an accumulation of MEPs showing downregulation of *Tal1*. This can be explained by a defect in the differentiation of these progenitors, and the phenotype we observe is reminiscent of the adult-specific *Tal1* knockout (Hall *et al*, 2005). It is therefore tempting to speculate that the regulatory circuit acting in the early embryo can be reenacted in the adult solely by induction of *Nanog*.

Hematopoietic differentiation of *Nanog^−/−* ES cells (Chambers *et al*, 2007) confirms the proposed role for *Nanog* in erythroid development. Although *Nanog^−/−* cells show an initial delay in the activation of early pan-mesodermal markers such as *Brachyury*, once this occurs, they show a faster and more coherent expression of erythroid genes. Directed differentiation reveals that the lack of *Nanog* promotes the red blood cell potential of these cells, which show a marked increase in both primitive and more mature erythroid colony formation. Our results show that *Nanog* acts as a barrier to red blood cell development. Controlled downregulation of *Nanog* during the initial phases of differentiation may present a novel approach to boosting the generation of red blood cells from pluripotent stem cells, a major clinical need (Kaufman, 2009).

# Materials and Methods

### Animal model

We obtained the *Nanog/rtTA* mouse line (*R26-M2rtTA;Col1a1-tetO-Nanog*) (Piazzolla *et al*, 2014) from Manuel Serrano (CNIO, Madrid) and Konrad Hochedlinger (Harvard Stem Cell Institute). This is a double transgenic line that carries the *M2-rtTA* gene inserted at the *Rosa26* locus and a cassette containing *Nanog* cDNA under the control of a doxycycline-responsive promoter (tetO) inserted downstream of the *Col1a1* locus. Mice were genotyped by PCR of tail-tip DNA as previously described (Hochedlinger *et al* 2005; Piazzolla *et al*, 2014). Mice were housed and maintained in the animal facility at the Centro Nacional de Investigaciones Cardiovasculares (Madrid, Spain) in accordance with national and European Legislation. Procedures were approved by the CNIC Animal Welfare Ethics Committee and by the Area of Animal Protection of the Regional Government of Madrid (ref. PROEX 196/14).

Double-homozygote transgenic males were mated with CD1 females, which were then treated with doxycycline (dox) to induce the *Nanog* cassette by replacing normal drinking water with a 7.5% sucrose solution containing dox (1 mg/ml), with replacement with fresh solution after 2 days. For transgene induction in embryos to

be harvested at E7.5, a single 100 μl intraperitoneal injection of 25 μg/μl doxycycline was administered to pregnant females at E5.5, followed by dox administration in drinking water as above.

### RT–qPCR assays

RNA was isolated from ESCs or sorted E9.5 cells using the RNeasy Mini Kit (Qiagen) and then reverse transcribed using the High Capacity cDNA Reverse Transcription Kit (Applied Biosystems). RNA from individual E6.5-7.5 embryos or sorted bone marrow populations was isolated using the Arcturus PicoPure RNA Isolation Kit (Applied Biosystems) and reverse transcribed using the Quantitect Kit (Qiagen).

cDNA was used for quantitative PCR (qPCR) with Power SYBR® Green (Applied Biosystems) in a 7900HT Fast Real-Time PCR System (Applied Biosystems). Expression of each gene was normalized to the expression of the housekeeping genes *Actin* and *Ywhaz*. Primers used are listed in Dataset EV2.

### Flow cytometry

E9.5 and E10.5 whole embryos or dissected yolk sacs were disaggregated with 0.25% collagenase type I (Stemcell Technologies) at 37°C for 30 min, and the cells were washed with PBS containing 2% FBS (Gibco) and filtered through a 70-μm mesh. The single-cell suspension was then incubated for 30 min at 4°C with the following antibodies: anti-CD71-FITC (BD Biosciences), anti-Ter119-APC (BD Biosciences), anti-cKit-PEcy7 (BD Biosciences), and anti-CD41-PE (BD Biosciences). Samples were analyzed with the BD LSRFortessa flow cytometer.

Bone marrow of adult mice was obtained from femurs and tibias crushed in a mortar and filtered through a 70-μm mesh to obtain single-cell suspensions. For hematopoietic cell maturation assays, a small fraction of the bone marrow was separated and the rest was depleted of red blood cells by lysis in FACSLysing solution (BD Biosciences). Antibodies used for blood maturation assay were anti-CD71-FITC (BD Biosciences) and anti-Ter119-APC (BD Biosciences). Antibodies for BM precursor sorting were Biotinylated lineage cocktail (BD Biosciences), anti-CD34(RAM34)-FITC (BD Biosciences), anti-cKit-PEcy7 (BD Biosciences), anti-CD16/32-BV605 (BD Biosciences), and anti-Sca1-PerCP-Cy5.5 (BD Biosciences).

### Cytospin cell preparation

For peripheral blood cytospin preparations, E9.5 embryos were dissected in warm PBS with 2% FBS and EDTA 0.5 mM, puncturing the yolk sac and the heart to let blood disperse into the media. All the preparation was passed through a 70-μm filter, centrifuged for 5 min at 135 *g*, and resuspended in a final volume of 200 μl PBS. Cells were collected on a glass slide on a Thermo ScientificCytospin 4 Cytocentrifuge for 10 min at 200 rpm and stained with May-Grünwald-Giemsa. Slides were scanned on a NanoZoomer-2.0RS C110730 scanner (Hamamatsu).

### Cell culture

ESCs were maintained in serum-free conditions with Knock out serum replacement (Thermo Fisher), LIF (produced in-house), and

2i (CHIR-99021, Selleckchem; and PD0325901, Axon). BT12 and E14Tg2a ESCs were kindly provided by Ian Chambers and Austin Smith (Chambers *et al*, 2007). ESC was differentiated toward hematopoiesis according to published protocols (Sroczynska *et al*, 2009; Irion *et al*, 2010; Lesinski *et al*, 2012).

For embryoid body formation, 5000 ESCs were plated in StemPro34 medium supplemented with nutrient supplement (Gibco) and 2 mM l-glutamine (l-Gln), penicillin/streptomycin (Gibco), 50 μg/ml ascorbic acid, 200 μg/ml iron-saturated transferrin, 4 ng/ml recombinant human BMP4, and $4 \times 10^{-4}$ monothioglycerol. After 2.5 days, to the cultures were added 5 ng/ml recombinant human fibroblast growth factor 2 (rhFGF2; basic fibroblast growth factor [bFGF]), 5 ng/ml recombinant human activin A, 5 ng/ml recombinant human VEGF (rhVEGF), 20 ng/ml recombinant murine thrombopoietin (TPO), and 100 ng/ml recombinant murine stem cell factor (rmSCF). Cytokines were obtained from R&D Systems Inc. or Peprotech. EBs were dissociated at day 6 by treatment with 0.05% trypsin-EDTA at 37°C for 2–5 min.

Dissociated EBs at day 5 and 6 were plated in Methocult SF M3436 methylcellulose medium for quantification of primitive erythroid progenitor cells (BFU-E). Dissociated EBs at days 5, 6, and 7 were plated in Methocult GF M3434 methylcellulose medium for quantification of erythroid progenitor cells (CFU-E), granulocyte–macrophage progenitor cells (CFU-GM, CFU-G, CFU-M), and multi-potential granulocyte, erythroid, macrophage, and megakaryocyte progenitor cells (CFU-GEMM). Cells were plated in triplicate on ultra-low attachment surface plates (Corning) at 50,000 cells per plate. Plates were incubated in high humidity chambers for 12 days at 37°C and 5% $CO_2$. Whole plates were counted. For qPCR, EBs were directly lysed in extraction buffer and frozen at −80°C.

*Nanog*-floxed ES cells (*Nanog^{flox/−}*; Zhang *et al*, 2018) were transfected with a Cre-expressing plasmid to induce recombination using Lipofectamine 2000 (Invitrogen). After 48 h, GFP-positive cells (*Nanog^{del/−}*) and GFP-negative cells used as control (*Nanog^{flox/−}*) were sorted using a FACS Aria Cell Sorter. Differentiation toward EpiLCs was induced by plating $5 \times 10^4$ ES cells on a well of a 24-well plate coated with human plasma fibronectin (10 μg/ml, Sigma) in N2B27 medium supplemented with 20 ng/ml Activin A (Preprotech),12 ng/ml bFGF (Preprotech), and 1% Knock out serum replacement (Thermo Fisher) for 3 days.

Embryonic stem cells from *Nanog^{tg}* mice were derived following standard procedures (Nagy *et al*, 2003). Differentiation to EpiLCs was performed in *Nanog^{tg}* ES cells and in two different clones of *Nanog^{tg}* ES cells where the binding site upstream of *Tal1* was deleted (*Nanog^{tg}*; *dTal1^{del #1}* and *^{#2}*). Differentiation was induced by plating $3 \times 10^4$ ES cells on a well of a 24-well plate and using the same conditions above-mentioned. After 3 days of differentiation, doxycycline (2 ng/ml) was added to the medium of the corresponding wells to induce *Nanog* expression. One day later, EpiLCs with or without doxycycline treatment were lysed for RNA isolation.

### *In situ* hybridization and immunohistochemsitry

Embryos were collected in cold PBS, transferred to 4% PFA, and fixed overnight at 4°C. After washing, embryos were dehydrated through increasing concentrations of PBS-diluted methanol (25, 50, 75, and 2× 100%). *In situ* hybridization in whole-mount embryos was performed as described (Ariza-McNaughton & Krumlauf, 2002;

Acloque *et al*, 2008). Signal was developed with anti-digoxigenin-AP (Roche) and BM-Purple (Roche). Images were acquired with a Leica MZ-12 dissecting microscope. Probes for *in situ* were obtained by PCR of cDNA with the primers listed in Dataset EV2.

For immunohistochemistry in whole mount, embryos were fixed overnight at 4°C in 4% paraformaldehyde, followed by overnight incubation at 4°C in primary antibody diluted 1:100 (rat monoclonal anti-endomucin, Santa Cruz sc-65495; or rat monoclonal anti-CD31, Santa Cruz sc-18916), washed and incubated overnight at 4°C with 1:500 Alexa Fluor 488 goat anti-rat (Termo Fisher Scientific, A-11006) for Endomucin or HRP goat anti-rat (Termo Fisher Scientific, 31470) for CD31. For histology, embryos fixed as above were dehydrated through an ethanol series, cleared with xylene, embedded in paraffin, sectioned at 5 μm, and stained with hematoxylin and eosin.

### RNA-seq

RNA was isolated from three replicates each of approximately 20,000 MEPs purified by sorting from adult untreated and dox-treated *Nanog*[tg] mice. Sequencing was performed by the CNIC Genomics Unit using the GAIIx sequencer. Adapters were removed with Cutadapt v1.14 and sequences were mapped and quantified using RSEM v1.2.20 to the transcriptome set from Mouse Genome Reference NCBIM37 and Ensembl Gene Build version 67. Differentially expressed genes between the two groups were normalized and identified using the limma bioconductor package. Only $P$-values < 0.05 adjusted through Benjamini–Hochberg procedure were considered as significant. Hierarchical clustering was performed on Z-scored values of the selected genes to generate an overview of the expression profile. Functional enrichment analysis was conducted using Enrichr (Kuleshov *et al*, 2016).

### CRISPR/Cas9 genome editing

sgRNAs were designed using the CRISPR Design Tool from the Zhang Lab at MIT (http://crispr.mit.edu/). Sequences of guide RNAs are indicated in Fig EV5A. The two guide RNAs at 25 ng/μl were incubated with the Cas9 protein (PNA bio) at 30 ng/μl and microinjected into the pronuclei of (CBAxC57)F1 zygotes (1,490); 1,075 surviving embryos were transferred to CD1 pseudopregnant females. 105 embryos were recovered at E6.5, and after discarding delayed or malformed embryos, 72 were lysed in 100 μl extraction buffer from the Arcturus PicoPure RNA Isolation Kit (Applied Biosystems). Aliquots of 10 μl were used for DNA extraction for PCR genotyping, and the remaining 90 μl was used for RNA extraction for RT–qPCR. Embryos for which we did not obtain a clear genotype were discarded, as well as those for which RT–qPCR of housekeeping genes did not reach a minimal threshold.

Embryonic stem cells from *Nanog*[tg] mice were electroporated with Cas9 protein and sgRNAs as above. Individual clones were picked, genotyped as above, karyotyped, and expanded for further use.

### Statistical analysis

Statistical analysis was performed with the use of two-tailed Student's unpaired *t*-test analysis (when the statistical significance of differences between two groups was assessed) or one-way ANOVAs with subsequent Fisher post-test (when the statistical significance of differences between more than two groups was assessed). Prism software version 7.0 (Graphpad Inc.) was used. For the analysis of the expected proportion of co-expressing cells with *Nanog,* we used a hypergeometric test in R.

## Data availability

Sequencing data have been deposited at GEO under accession number GSE119467 (https://www.ncbi.nlm.nih.gov/geo/query/acc.cgi?acc=GSE119467).

**Expanded View** for this article is available online.

## Acknowledgements

We thank Manuel Serrano and Konrad Hochedlinger for the *Nanog*[tg] mouse line; Miguel Torres and Covadonga Díaz for the ES-GFP cell line; Austin Smith, Ian Chambers, and Harry G. Leitch for *Nanog*[−/−] ES cell lines; Luis Miguel Criado and the CNIC Transgenesis Unit for chimera generation; Elena Lopez-Jimenez, Giovanna Giovinazzo, and the CNIC Pluripotent Cell Technology Unit for derivation of *Nanog*[tg] ES cells; Simon Mendez-Ferrer and Abel Sánchez-Aguilera for support and discussions; Cristina Gutierrez-Vazquez, Teresa Rayon, Hector Sanchez-Iranzo, and Andrés Hidalgo for comments on the manuscript; Simon Bartlett for English editing; and members of Manzanares laboratory for continued support. This work was supported by the Spanish government (grant BFU2014-54608-P and BFU2017-84914-P to MM; grants RYC-2011-09209 and BFU-2012-35892 to JI). The Gottgens and Nichols laboratories are supported by core funding from the Wellcome Trust and MRC to the Wellcome and MRC Cambridge Stem Cell Institute. The CNIC is supported by the Spanish Ministry of Science, Innovation and Universities (MINECO) and the Pro CNIC Foundation, and is a Severo Ochoa Center of Excellence (SEV-2015-0505).

## Author contribution

JSA and MM designed the research with input from JI, BG, and JN. JSA performed the majority of the experiments, with the help of SM for embryo work and tissue culture; IR for genome editing; AB and MT for ES cell work; WJ and AA for data analysis; IC, AB, and GC-T for analysis of adults; and CB-C for in situs. JSA and MM wrote the manuscript with input from all authors.

## Conflict of interest

The authors declare that they have no conflict of interest.

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
