## [Review Process File · The EMBO Journal]

The pluripotency factor NANOG controls primitive hematopoiesis and directly regulates *Tal1*

Julio Sainz de Aja, Sergio Menchero, Isabel Rollan, Antonio Barral, Maria Tiana, Wajid Jawaaid, Itziar Cossio, Alba Alvarez, Gonzalo Carreño-Tarragona, Claudio Badia-Careaga, Jennifer Nichols, Berthold Göttgens, Joan Isern and Miguel Manzanares

Review timeline:

Submission date:	30th Jan 2018
Editorial Decision:	21st Mar 2018
Revision received:	7th Dec 2018
Editorial Decision:	17th Jan 2019
Revision received:	24th Jan 2019
Accepted:	25th Jan 2019

Editor: Daniel Klimmeck

Transaction Report:

1st Editorial Decision

21st Mar 2018

Thank you again for the submission of your manuscript (EMBOJ-2018-99122) to The EMBO Journal and in addition providing us with a preliminary revision plan. Thank you also for your patience with my response, which got delayed due to detailed discussions in the team, as well as getting back to the referees regarding your preliminary point-by-point response. As mentioned earlier, your study has been sent to three referees, and we have received reports from all of them, which I copy below.

The referees acknowledge the potential interest and novelty of your work, although they also express major concerns. While referee #1 is overall more positive, referee #2 raises reservations regarding the *in vivo* physiological relevance of Nanog in embryonic erythropoiesis, and states that the claims made are not sufficiently supported by the data. In addition, this referee states that the evidence for control of Scl-Tal1 by Nanog is unconvincing in his-her view as to the subtlety of changes in the data. Referee #3 agrees in that the direct relevance of the proposed Nanog-Tal1 axis for the phenotypes observed remains unclear.

We realise that you would - judging from the information provided in the point-by-point letter - be potentially able to address the issues raised by the referees in a revised version of the manuscript. This view was shared by the referees who stated that your suggested revision experiments would considerably strengthen the study. Please see below for additional comments.

I judge the comments of the referees to be generally reasonable and can - based on your sensible preliminary response - offer to invite you to revise your manuscript along the experiments outlined in your plan to address the referees' concerns. I agree that in particular the aspect of physiological relevance of Nanog in embryonic erythropoiesis and control of Scl-Tal1 by Nanog would need to be conclusively addressed in a revised version of the manuscript to move towards publication.

REFEREE COMMENTS:

Referee #1:

The manuscript by Sainz de Aja et al reports the results of studies examining the influence of Nanog expression on erythroid differentiation. The authors test the hypothesis that Nanog is a direct regulator of Tal expression and erythropoiesis using Tet-inducible in vivo and in vitro and knockout approaches during mouse embryonic and adult stages. ES cell chimera studies show that the affect is cell autonomous and the wise use of published databases allow for the important conclusion that Nanog binds upstream of the Tal1 gene. Deletion of this binding site provides confirmation of that Nanog is a suppressor of Tal1 expression in erythroid differentiation. These results are of interest to the general hematopoietic as well developmental biology communities and could have impact on the reprogramming or induced differentiation of non-hematopoietic cells to the erythroid lineage.

Referee #2:

The authors have shown that inducing Nanog during embryonic development messes up early erythropoiesis. While this definitely tells us that this pluripotency factor can disrupt the early hematopoietic program, it is not surprising, and I am not convinced by this data that negatively regulating erythropoiesis is an important biological function for Nanog. Functional necessity can only be determined by loss of function experiments. The only Nanog loss of function experiment in this study involves in vitro differentiation of ES cells, uses only a single line of Nanog mutant cells compared to a single line of WT cells. From experience, different established lines of ES cells have vastly different efficiencies as well as timings of hematopoietic progenitor formation. Thus the differences might be clonal variation. It is simply not convincing. Short of conditional deletion of Nanog in the progenitors of this lineage during embryonic development, one cannot claim a role for Nanog in regulating this process. Unfortunately, this fundamental problem makes it likely that this entire study is exploring a gain of function artifact. One could imagine that many genes, if overexpressed, might lead to interference with particular lineage-specific pathways during development. As such, the study would be better suited to a more archival journal.

The identification of a putative distant regulatory site for SCL is potentially interesting. However the data is extremely weak, and very preliminary. Why the authors did this in the much less controllable system of embryo development when if their reagent works, they could easily have derived ES lines with this element deleted and studied them using in vitro differentiation, and done the experiment with a sufficient number of data points and replicates to make it convincing is unclear to me. Injection of materials into embryos can interfere with later development. It is quite likely that the embryos that were not modified were somehow injected with less or simply unrealized injection failures. Thus, the difference in SCL expression could simply be due to differences in modest interference with embryonic development due to differences in injection. Also, the assay of RTPCR on whole embryos is an extremely crude assay. To rigorously establish that this regulatory element is essential, the authors should:

1. Derive multiple deleted ES cells lines, and not-deleted control lines, differentiate these and evaluate efficiency of SCL expression and hematopoietic CFC differentiation. Repeat the analysis on the panel of clones at least twice to ensure that it is replicable.
2. Perform the Cas9 deletion in inducible Nanog embryos, rather than in WT embryos. With deletion of the regulatory element, inducible Nanog should no longer be able to misregulate hematopoiesis.
3. Compare to unmanipulated embryo controls. Show levels of expression of other hematopoietic genes, to show that the effect is specific to SCL.

Additional suggestions for the authors:

Most of the experiments appear to have been performed only once, at least replicates are not mentioned. For example, the experiments in Figure 5G, evidence that deletion of the putative SCL regulatory element causes a change in SCL expression levels provides only that n=5, presumably 5

mutant and 5 WT embryos from a single experiment. The variance is large and the data points are not shown, just the average. Was this experiment repeated, or is it a one-off?

This same criticism applies throughout the paper. Whether experiments have been replicated or are one-offs is generally not mentioned.

Some of the data is presented as scatter plots, while other data is presented as bar graphs showing mean. The bar graphs needlessly hide the distribution, as well as the small group sizes. Individual data points should be shown for all graphs, not just those that have larger sample sizes.

The authors do not cite the appropriate reference for the claim that CD41 marks the earliest hematopoietic progenitors.

Referee #3:

General summary:

This manuscript addresses the mechanisms that control the specification primitive erythroid lineage from multipotent mesodermal precursors. The authors identify new function for Nanog, one of the pluripotency regulators, in the epiblast, in repressing premature differentiation to the erythroid lineage. Overexpression of Nanog beyond its normal time of expression using doxycycline inducible mouse model results in compromised primitive erythroid differentiation in midgestation embryos. No obvious defect is observed in endothelium, and CD41+cKit+ hematopoietic progenitors in the whole embryo are increased. Chimera analysis mixing WT-GFP ES cells with Dox-inducible Nanog blastocyst show the superior contribution WT ES cell to the erythroid cells in the embryo. In contrast, Nanog deficient ES cells showed an opposite effect, ie. increased differentiation to primitive and definitive erythroid cells, as measured by Ery-P and BFU-E colonies derived from the ES cells, while mixed colonies were reduced. Surprisingly, the overall Tal1 level was not affected in the ES cell derived differentiated cells, although there was a delay in downregulation of mesodermal marker, Brachyury. Induction of Nanog expression in post-natal life using the dox-inducible mouse model results in increase in MEPs in the BM, as well as impaired differentiation of BM erythroid cells beyond the erythroblast stage. ChIP-seq analysis suggests that Tal1 is a direct target of Nanog in Epiblast stage. CRISPR-Cas9 deletion of the Nanog bound candidate Tal1 regulatory region results in twofold increase in Tal1 expression without affecting Klf1 expression.

This study is interesting as it focuses on a poorly understood process of how mesoderm generates the primitive erythroid lineage. A lot of the recent focus has been on mechanisms that govern the specification of hemogenic endothelium from mesoderm, whereas this direct specification step from mesoderm to the first hematopoietic cells in the embryo has been largely ignored. Although previous studies had shown that Nanog deficient ES cells show promiscuous differentiation to the erythroid lineage, however, this manuscript shows this for the first time in an in vivo setting, using a combination of loss and gain of function mouse models. Moreover, this study is set up nicely to study this process mechanistically. However, although the data is potentially interesting, some of the experimental design and data analysis are done in "unconventional" way and require additional studies and data documentation to be fully evaluated. See specific comments below:

Major concerns:

- Although Tal1 seems to be regulated by Nanog, given the fairly modest expression changes of Tal1 that are not always consistent, it is not clear if Tal1 regulation is the major driver of the observed phenotypes. Performing genome wide gene expression rather than limiting the analysis to RT-PCR of candidate regulators could provide more complete picture of the key downstream targets and programs that could be driving the phenotypes.

- Minimal data is shown of the embryos in which Tal1 regulatory element is removed.

Other concerns and suggestions:

- The rationale for using Redrum lncRNA as an erythroid marker is not clear, as the AGM and tail bud where their expression is reported by the authors are not sites of erythropoiesis. Is it expressed in AGM endothelium, or in circulating erythroid cells found in the aorta? Please also provide a reference for the expression pattern of Redrum.

- As some primitive erythroid cells are present, it would be helpful to perform MGG or other staining to show their morphology. Are they just reduced in number, or also blocked in differentiation?
- The authors show that vascular development appears unaffected, but there are more CD41+ckit+ hematopoietic progenitors present. Is this an absolute or relative increase in the embryo? There is minimal discussion about these results and the increase in Runx1 expression. It is also difficult to fully interpret these results as whole embryo tissue was analyzed rather than dissection specific hematopoietic organs, most importantly the yolk sac.
- Tal1 expression is initially assessed in the small subpopulation of CD71+ter119+ cells that can develop, and the decrease appears modest. One could assume that more drastic downregulation could potentially be observed at earlier Flk1 mesoderm. This appears to be the case based on Fig 5 where more complete downregulation of Tal1 is observed when the early embryos were analyzed as a whole. However, Tal1 is not only erythroid cell specifier, but its complete loss leads to loss of CD41+ckit+ cells in the yolk sac and embryo. If Tal1 expression in mesoderm is repressed by Nanog, why is there increase in CD41+ckit+ cells? Can the authors comment on these discrepancies?
- A comment "Gain of Nanog expression in mature erythrocytes (CD71+Ter119+ population) resulted in significant downregulation of Tal1 and upregulation Runx1" is not completely correct as they still retain CD71 expression and nucleus. Mature erythrocytes, even primitive, are identified by enucleation.
- It would be helpful to show vascular staining also in the Yolk sac to provide evidence that vascular development in hematopoietic sites is normal as their regulation in hemogenic vasculature could be different.
- As a general phenomenon, the figures are not very informative alone as they lack information about tissues used for the analysis, embryonic age etc. Adding some detail would make it easier to follow both for reviewers and readers, as figure legends are not always available in the same page.
- Does the heart beat normally? Given that heart develops from Flk1 mesoderm similar to blood and vascular lineages, and proposed Nanog target Tal1 is known to repress cardiac fate, it would be good to give a general idea whether the heart development is generally unaffected.
- The reduction in Tal1 levels in the BM erythroblasts does not appear very drastic and unlikely to explain the defect as a whole, as Tal1 heterozygous mice do not have obvious erythroid defect. The accumulation of CD71+Ter119+ erythroblast and minimal evidence of enucleated erythroid cells is a strong phenotype, but it is unclear how this is caused. Genome wide gene expression profiling would likely provide a more comprehensive picture and might identify programs that are not even considered otherwise.
- EV.1 Increase of Tal1 in ckit-CD41+ cells in Dox treated embryos is difficult to interpret as it is not clear what cells are included in this population. But these results generally would argue against Tal1 being a main direct target that is repressed by Nanog.
- EV4 gel needs labeling

EMBOJ-2018-99122

Sainz de Aja et al.

Response to referees

In first place, we wish to thank the editor and the referees for acknowledging the potential interest and novelty of our work and for the in-depth revision of our manuscript. We have studied carefully all concerns raised by the referees, and we believe we have positively address most if not all of the issues raised. Below follows a point-by-point response to the referees, and apart from minor changes the following new data has been added to the manuscript:

- Quantification of hematopoietic progenitors from E9.5 dissected yolk sacs, instead of whole embryos.
- Analysis of the yolk-sac vasculature and the cardiac phenotype in *Nanog*-expressing embryos.
- Analysis of the effect of the loss of *Nanog* on the hematopoietic gene program using an independent mutant ES cell line.
- Genome-wide expression analysis of adult progenitors (MEP) to better characterize the *Nanog*-driven phenotype.
- More complete characterization of the adult phenotype, including results from bone-marrow transplantation of *Nanog*^{tg} precursors.
- Increase in the number of in vivo edited embryos for the deletion of the *Tal1* regulatory element and analysis of other hematopoietic genes.
- Analysis of ES cells deleted for the *Tal1* regulatory element.

These new experiments, together with a profound revision of the text and figures as suggested by the referees, has led to a much improved version of the manuscript.

Point-by-point response

Referee #1:

The manuscript by Sainz de Aja et al reports the results of studies examining the influence of *Nanog* expression on erythroid differentiation. The authors test the hypothesis that *Nanog* is a direct regulator of *Tal* expression and erythropoiesis using Tet-inducible in vivo and in vitro and knockout approaches during mouse embryonic and adult stages. ES cell chimera studies show that the affect is cell autonomous and the wise use of published databases allow for the important conclusion that *Nanog* binds upstream of the *Tal1* gene. Deletion of this binding site provides confirmation of that *Nanog* is a suppressor of *Tal1* expression in erythroid differentiation. These results are of interest to the general hematopoietic as well developmental biology communities

and could have impact on the reprogramming or induced differentiation of non-hematopoietic cells to the erythroid lineage.

We are very grateful to the referee for the very encouraging comments on our work. We hope that the revised version will further support this positive view.

Referee #2

The authors have shown that inducing *Nanog* during embryonic development messes up early erythropoiesis. While this definitely tells us that this pluripotency factor can disrupt the early hematopoietic program, it is not surprising, and I am not convinced by this data that negatively regulating erythropoiesis is an important biological function for *Nanog*. Functional necessity can only be determined by loss of function experiments. The only *Nanog* loss of function experiment in this study involves in vitro differentiation of ES cells, uses only a single line of *Nanog* mutant cells compared to a single line of WT cells. From experience, different established lines of ES cells have vastly different efficiencies as well as timings of hematopoietic progenitor formation. Thus the differences might be clonal variation. It is simply not convincing.

We agree with the referee of the importance of demonstrating the proposed role of *Nanog* by loss of function. Because of the early embryonic lethality of the full *Nanog* KO (Mitsui et al., Cell 2003), we decided to use mutant ES cells as a model system to analyze the role of *Nanog* in for hematopoietic gene expression and differentiation. Certainly, in this experimental context and as the referee points out, variability in differentiation capabilities of different ES cell lines can be a concern. Therefore, we have tried to control this experiment as much as possible. We obtained the *Nanog* KO ES cell line from Austin Smith and Ian Chambers, who also provided us with the parental wild type line from which the mutant *Nanog* ES cell line was derived which we have used as control (Chambers et al., Nature 2007). We are not aware of clonal variability of this particular ES cell line, but in order to address this concern, we obtained an independently generated *Nanog* mutant ES cell line from Harry G. Leitch (Zhang et al., Cell Rep 2018).

This cell line is maintained as a heterozygote, with a *Nanog* floxed allele and a *Nanog* deleted allele. Upon Cre activity, the floxed allele is removed activating GFP expression, and thus generating a *Nanog* null ES cell. Our first intention was to generate stable *Nanog*^{del/-} ES cell line to test changes to their hematopoietic potential and gene expression profile upon differentiation. However, and despite numerous attempts of both our group and the Pluripotent Cell Unit of our institute, we have been unable to establish the mutant line after Cre transfection. Deleted cells (as detected by GFP expression) quickly differentiate to a primitive endoderm-like phenotype and die if maintained under ES cell culture conditions (both 2i+LIF and LIF+serum). These unfortunate circumstances have prevented us from obtaining sufficient mutant cells to carry out a properly scaled differentiation experiment.

To circumvent this issue, we have adopted an alternative strategy, where we grew heterozygous ES cells, transfected them with Cre, sorted for GFP, and cultured en masse to differentiate them to epiblast-like cells (EpiLC; Hayashi et al., Cell 2011). This process recapitulates in culture the transition from pluripotent cells of the blastocyst to primed cells of the epiblast (Buecker et al., Cell Stem Cell 2014). *Nanog* is expressed

throughout this time window, and it is during this stage when we hypothesize it is having a role in regulating the differentiation of erythroid precursors. We could not use this protocol for differentiation to embryoid bodies because of the small amount of sorted GFP+ cells that are obtained.

We compared gene expression along differentiation to EpiLCs of GFP+ (*Nanog*^{del/-}) and control GFP- cells (*Nanog*^{flox/-}), finding that *Nanog* mutant cells show a precocious expression of hematopoietic markers such as *Tal1*, *Gata1* or *Klf1*. These results confirm our previous observations in a different ES cell line. Furthermore, this is a nice confirmatory experiment, as in this case we observe the effect of the acute loss of *Nanog* at the precise time point when ES cells start to differentiate, while in the case of BT12 cells these have been continuously going in the absence of *Nanog*. We have included the new data in Fig EV2C, and described the results in the text.

Short of conditional deletion of *Nanog* in the progenitors of this lineage during embryonic development, one cannot claim a role for *Nanog* in regulating this process. Unfortunately, this fundamental problem makes it likely that this entire study is exploring a gain of function artifact. One could imagine that many genes, if overexpressed, might lead to interference with particular lineage-specific pathways during development. As such, the study would be better suited to a more archival journal.

We agree with the referee that conditional deletion of *Nanog* in hematopoietic precursor, following a similar strategy to that recently published (Zhang et al., Cell Rep 2018), would provide very useful information regarding the in vivo role for *Nanog* at postimplantation stages. Unfortunately, we do not have access to this mouse line, and the time needed to generate a similar model exceeds what could be reasonable to complete the present manuscript.

Another concern of the referee is that we are observing a non-specific effect, and that overexpression of other genes could lead to similar defects. Certainly, this could be the case, but we believe that we provide sufficient evidence to prove that *Nanog* is acting specifically on early hematopoiesis. In first place, *Nanog* expressing embryos show a great reduction in red blood cells. We observe this phenotype at E9.5 (Fig 1A) as well

Fig R1. A, *Nanog* reduces red blood cells in E10.5 and E14.5, embryos after 3-days of dox treatment. **B**, *Oct4* expression up to E10.5 does not lead to loss of red blood cells.

as at other developmental times points (E10.5, E14.5; Fig R1A). In second place, in the lab we are also using an *Oct4* gain-of-function model in a similar fashion (Lopez-Jimenez et al., under review), and in this case, although embryos show clear morphological abnormalities at E10.5, they produce blood (Fig R1B). Therefore, at least compared to another core pluripotency factor, the effect of *Nanog* on erythropoiesis is specific.

Finally, the effect we observe of *Nanog* on hematopoietic regulators is not secondary to a general disruption of embryonic development. For example, it would be easy to imagine that gain of function of *Nanog* or other pluripotency genes would lead to shutting off early mesoderm lineages and secondarily no blood development. However, we do not observe so, and even more, at E7.5 *Nanog* is capable of inducing key early pre-mesoderm specifiers such as *Brachyury*, a known inducer of primitive erythropoiesis (Fehling et al., Development 2003), or *Eomes*, but nevertheless downregulates erythroid genes (Fig EV1F).

We should also stress that we are using the transgenic gain of function model as a discovery tool to interrogate putative roles for *Nanog* in the postimplantation embryo. Furthermore, we are not in any way assuming that the only role of *Nanog* at these stages is to regulate the first wave of hematopoiesis. In fact, we are also exploring how *Nanog* represses anterior neural fates (Sainz de Aja et al., in preparation), what explains the craniofacial defects we observe in the embryos (Fig 1A).

The identification of a putative distant regulatory site for SCL is potentially interesting. However the data is extremely weak, and very preliminary. Why the authors did this in the much less controllable system of embryo development when if their reagent works, they could easily have derived ES lines with this element deleted and studied them using in vitro differentiation, and done the experiment with a sufficient number of data points and replicates to make it convincing is unclear to me.

We decided to perform the deletion of the regulatory element directly in the embryo because we were interested in addressing the endogenous role of *Nanog* in vivo during early development. We agree with the referee that the initial number of analyzed embryos (5) is low. We have now increased this number to 19 non-deleted and 13 deleted E6.5 embryos analyzed for *Tal1* and *Stil* expression (revised Fig 5G).

Nevertheless, the in vitro approach in ES cells suggested by the referee is of course also of great interest, and we have generated the lines for this purpose (see below).

Injection of materials into embryos can interfere with later development. It is quite likely that the embryos that were not modified were somehow injected with less or simply unrealized injection failures. Thus, the difference in SCL expression could simply be due to differences in modest interference with embryonic development due to differences in injection. Also, the assay of RTPCR on whole embryos is an extremely crude assay.

The referee offers a plausible explanation for our results, but we do believe that the way we carry out these experiments precludes this possibility. Reading the original

manuscript, we can see that the description of the experiment is not very clear. We are sorry for this, and have corrected the manuscript accordingly. Briefly, for this set of experiments, we injected 1490 embryos at different days with the Cas9/gRNA ribonucleoprotein complexes, of which 1075 survived and were transferred to pseudo-pregnant recipient females. We recovered 105 embryos at E6.5, but at this point we only kept embryos that were morphologically normal (72 embryos), discarding any delayed or malformed embryos. Only then, we extracted DNA and genotyped the stage-matched embryos for the deletion. Embryos from which we do not obtain a clear genotype (for example, see Fig EV5B) were also discarded, keeping 50 embryos for RT-qPCR. Those cases in which housekeeping genes did not reach a minimal threshold were also removed from the analysis, leaving a final number of 32 embryos to include in the analysis (19 non-deleted and 13 deleted). Although we have only examined a small number of genes (due to limitations in the material we can obtain from individual E6.5 embryos, and taking into account that we also need to genotype them), we only observe changes for *Tal1* and not for *Klf1*, *Gfi1b*, *Runx1* or *Stil* (see below and revised Fig 5G). If we were observing a general non-specific deregulation of development, we would not expect to see changes only in one of the genes examined.

The referee also mentions that quantitative RTPCR is an extremely crude assay. We can only think of whole mount in situ as an alternative to examine gene expression, but this approach does not allow a quantitative analysis of the result, only qualitative.

To rigorously establish that this regulatory element is essential, the authors should:

1. Derive multiple deleted ES cells lines, and not-deleted control lines, differentiate these and evaluate efficiency of SCL expression and hematopoietic CFC differentiation. Repeat the analysis on the panel of clones at least twice to ensure that it is replicable.

Following the referee's advice, and as mentioned above, we have generated ES cell lines with the *Tal1* regulatory element deleted in homozygosity. Furthermore, this has been done in the background of the double transgenic dox-inducible *Nanog* system. In this way, we have tested both the effect of the deletion in a normal situation (-dox) and also addressed if this deletion renders *Tal1* unresponsive to *Nanog* expression.

In order to address the effect of the deletion on *Tal1* expression and its dependence on *Nanog*, we differentiated ES cells to EpiLC, as we have described above. We observe that non-treated *Nanog*^{tg} ES cells (that would be our control) show a significant increase in *Tal1* expression when they transit to EpiLCs, what would be the equivalent of the initial expression of *Tal1* in the embryo. However, if dox is added to the medium, this increase of *Tal1* between ES and EpiLC is no longer significant. Thus, in this experimental setup, increased expression of *Nanog* is able to block at least partially the early induction of *Tal1*, in line with our in vivo results. Nevertheless, when we repeat his experiment but with two independent ES cell lines where the NANOG-bound distal element (*dTal1*) has been deleted, EpiLC become unresponsive to *Nanog* upon dox treatment and still upregulate *Tal1* as cells not treated with dox. These new results show that the distal element we have characterized is necessary for correct initiation of *Tal1* expression, and that it is the mediator of the response of *Tal1* to *Nanog*. As we do not believe that *Tal1* is the sole mediator of *Nanog*'s effect on erythroid cells (see above and response to Referee #3), and also that other positive regulatory inputs are acting on *Tal1* to trigger its expression, we would not expect to see an effect of the

deletion on later hematopoietic differentiation and colony forming capacities. We have added this new data to the manuscript (revised Fig 5H and Fig EV5C) and discussed it accordingly in the text.

2. Perform the Cas9 deletion in inducible *Nanog* embryos, rather than in WT embryos. With deletion of the regulatory element, inducible *Nanog* should no longer be able to misregulate hematopoiesis.

The experiment proposed by the referee would certainly provide further evidence for the direct regulation of *Tal1* by *Nanog* and the role of this regulatory interaction in early hematopoiesis. However, due to technical limitations we cannot carry out this experiment at present. In order to generate the deleted embryos described above, we work with a stable colony of CBAxC57 crosses to generate F1 hybrids to produce one-cell embryos for microinjection, following standard transgenic procedures. This represent a colony of over 200 cages. If we wanted to generate the deletions in the background of the inducible *Nanog* line, we would need to expand this particular colony in homozygosity to a size (minimal 100 cages) that is unfeasible due to the limitations in our mouse colony. An additional problem would be the well-known lower survival and overall transgenic efficiency when using particular mouse lines as compared to standard F1 hybrids.

However, in order to address this point, we have generated the deletion in ES cell lines derived from the double transgenics dox-inducible *Nanog* mice (see above). Certainly, we could use these ES cells to generate mouse lines. However, the period for the generation and analysis of this line (approximately 1-year) prevents us from incorporating the putative results in a revised version of the manuscript in a realistic time frame.

3. Compare to unmanipulated embryo controls. Show levels of expression of other hematopoietic genes, to show that the effect is specific to SCL.

In our view, embryos that have been manipulated in the same way as described above but do not show the deleted band by genotyping are the best-matched controls. The referee is concerned that micro-manipulated embryos showing a wild type genotype could be injection failures, and thus not showing developmental abnormalities related to the microinjection process. In this case, these would be the same as unmanipulated embryos.

Following the referee's suggestions, we have extended the panel of hematopoietic genes analyzed and included *Gfi1b* and *Runx1* (see revised Fig 5G). We have also increased the number of embryos in which we have analyzed *Klf1* and *Stil*, as for *Tal1* (see above). The lower number of embryos analyzed for *Klf1*, *Gfi1b* and *Runx1* is due to these genes being expressed at very low levels at E6.5, and therefore those cases where we did not detect expression were not included in the analysis. Nevertheless, we observe that none of these genes shows differential expression between deleted and control embryos, further confirming our conclusion that the deleted element is specifically regulating *Tal1*.

Additional suggestions for the authors:

Most of the experiments appear to have been performed only once, at least replicates are not mentioned. For example, the experiments in Figure 5G, evidence that deletion of the putative SCL regulatory element causes a change in SCL expression levels provides only that n=5, presumably 5 mutant and 5 WT embryos from a single experiment. The variance is large and the data points are not shown, just the average. Was this experiment repeated, or is it a one-off?

As we mention above, the deleted embryos analyzed in this experiment come from multiple microinjection sessions, as do the stage-matched controls. Because of the nature of transgenic embryo generation, each individual embryo can be considered as an independent experiment (CRISP/Cas9 mediated deletions will occur in an independent manner in each microinjected 1-cell embryo). It is true that variance is large, but we must also take into account that these deleted embryos will be mosaic (as evidenced by the genotyping shown in Fig. EV5B). Therefore, we assume that the observation of increased *Tal1* expression is an underestimate. We have followed the referee's suggestion, and now present data points and not averages in Fig 5G. We have also done so for all other graphs in the manuscript.

This same criticism applies throughout the paper. Whether experiments have been replicated or are one-offs is generally not mentioned.

As for this observation of the referee, we must have failed to explain ourselves properly in the manuscript, as all experiments have been performed multiple times. Numbers of replicates are indicated in the figure legends, and only in those cases where data points are shown in the figure we have not included this information (for example, Fig. 1D, E). To make this point clearer, we have revised the manuscript to include all this information where necessary.

Some of the data is presented as scatter plots, while other data is presented as bar graphs showing mean. The bar graphs needlessly hide the distribution, as well as the small group sizes. Individual data points should be shown for all graphs, not just those that have larger sample sizes.

As mentioned above, we have corrected all the figures to now show all of quantitative data as scatter plots.

The authors do not cite the appropriate reference for the claim that CD41 marks the earliest hematopoietic progenitors.

We are sorry for the mistake and we have now include the correct reference for this claim (Ferkowicz et al. Development 2003).

Referee #3

This manuscript addresses the mechanisms that control the specification primitive erythroid lineage from multipotent mesodermal precursors. The authors identify new function for Nanog, one of the pluripotency regulators, in the epiblast, in repressing premature differentiation to the erythroid lineage. Overexpression of Nanog beyond its normal time of expression using doxycycline inducible mouse model results in compromised primitive erythroid differentiation in midgestation embryos. No obvious defect is observed in endothelium, and CD41+ckit+ hematopoietic progenitors in the whole embryo are increased. Chimera analysis mixing WT-GFP ES cells with Dox-inducible Nanog blastocyst show the superior contribution WT ES cell to the erythroid cells in the embryo. In contrast, Nanog deficient ES cells showed an opposite effect, ie. increased differentiation to primitive and definitive erythroid cells, as measured by Ery-P and BFU-E colonies derived from the ES cells, while mixed colonies were reduced. Surprisingly, the overall Tal1 level was not affected in the ES cell derived differentiated cells, although there was a delay in downregulation of mesodermal marker, Brachyury. Induction of Nanog expression in post-natal life using the dox-inducible mouse model results in increase in MEPs in the BM, as well as impaired differentiation of BM erythroid cells beyond the erythroblast stage. ChIP-seq analysis suggests that Tal1 is a direct target of Nanog in Epiblast stage. CRISPR-Cas9 deletion of the Nanog bound candidate Tal1 regulatory region results in twofold increase in Tal1 expression without affecting Klf1 expression.

This study is interesting as it focuses on a poorly understood process of how mesoderm generates the primitive erythroid lineage. A lot of the recent focus has been on mechanisms that govern the specification of hemogenic endothelium from mesoderm, whereas this direct specification step from mesoderm to the first hematopoietic cells in the embryo has been largely ignored. Although previous studies had shown that Nanog deficient ES cells show promiscuous differentiation to the erythroid lineage, however, this manuscript shows this for the first time in an in vivo setting, using a combination of loss and gain of function mouse models. Moreover, this study is set up nicely to study this process mechanistically. However, although the data is potentially interesting, some of the experimental design and data analysis are done in "unconventional" way and require additional studies and data documentation to be fully evaluated. See specific comments below:

We thank the referee for the comments and for appreciating the novelty of our results and the importance of analyzing them in an in vivo setting.

Major concerns:

- Although Tal1 seems to be regulated by Nanog, given the fairly modest expression changes of Tal1 that are not always consistent, it is not clear if Tal1 regulation is the major driver of the observed phenotypes. Performing genome-wide gene expression rather than limiting the analysis to RT-PCR of candidate regulators could provide more complete picture of the key downstream targets and programs that could be driving the phenotypes.

As the referee rightly points out, we do not believe that the effects of Nanog on primitive hematopoiesis are only mediated by downregulation of Tal1. We do observe changes in other key regulators of this process. For example, Gata1 is also an interesting candidate to be a direct target of Nanog, as we observe it is downregulated

in E7.5 embryos (both by in situ and by RT-qPCR; revised Fig EV1E, F) and has EpiLC specific NANOG bound regions in its vicinity (Fig EV4). However, we have not pursued the characterization of this other potential direct link due to time and organizational constraints.

We have tried hard to avoid delivering the message that *Nanog* plays its role in early hematopoiesis just through *Tal1* downregulation. However, in light to the referee's comment and re-reading our original manuscript we can see that this is not clear enough. We have therefore modified the title of our manuscript as well as the discussion to make this point clear.

The suggestion to perform genome-wide gene expression analysis is of clear interest, as it could provide a view of how *Nanog* regulates distinct pathways related to the phenotype. To carry out this analysis, we decided to use adult MEPs, as this is a well defined cell type that we have shown is blocked in its differentiation capacity towards more mature erythroid cells by *Nanog*. The results from RNA-seq analysis of MEPs from untreated and dox-treated *Nanog^{tg}* mice is described below.

- Minimal data is shown of the embryos in which *Tal1* regulatory element is removed.

Again, we agree with the referee on this point, and have extended the analysis by generating more deleted embryos, analyzing the expression of further hematopoietic genes, and also by examining the result of deleting the *Tal1* regulatory element in ES cells derived from *Nanog^{tg}* mice that were differentiated to EpiLCs. Please see the above response to Referee #2 for a detailed description of the new data we have generated to address this issue.

Other concerns and suggestions:

- The rationale for using *Redrum* lncRNA as an erythroid marker is not clear, as the AGM and tail bud where their expression is reported by the authors are not sites of erythropoiesis. Is it expressed in AGM endothelium, or in circulating erythroid cells found in the aorta? Please also provide a reference for the expression pattern of *Redrum*.

We selected *Redrum* as it has been described as a lncRNA expressed in erythroblast cells (Alvarez-Dominguez et al., Blood 2014, Paralkar et al., Blood 2014), which at this stage can be seen circulating in the dorsal aorta. Therefore, we argue that lack of expression in the AGM region is due to loss of circulating *Redrum* expressing cells. However, other domains of *Redrum* expression, such as the tail bud that are not sites of erythropoiesis, are not affected by induction of *Nanog* expression. We have modified the text accordingly to provide a clearer description of our observations.

- As some primitive erythroid cells are present, it would be helpful to perform MGG or other staining to show their morphology. Are they just reduced in number, or also blocked in differentiation?

We appreciate this interesting suggestion from the referee, and have therefore isolated by cytopsin blood cells from E9.5 dox-treated and untreated embryos and stained them with May-Grünwald-Giemsa to show morphology. We see that overall the morphology

of the few remaining primitive erythroid cells from *Nanog*^{tg} E9.5 embryos treated with dox is fairly normal, with slightly less basophilic cytoplasm. We have included this data in the text and in Fig EV1C.

- The authors show that vascular development appears unaffected, but there are more CD41+ckit+ hematopoietic progenitors present. Is this an absolute or relative increase in the embryo? There is minimal discussion about these results and the increase in Runx1 expression. It is also difficult to fully interpret these results as whole embryo tissue was analyzed rather than dissection specific hematopoietic organs, most importantly the yolk sac.

We agree with the referee that the fact of using whole embryos in these experiments could obscure the data, and have therefore repeated the analysis using dissected yolk sacs from E9.5 dox-treated and untreated embryos (see revised Fig 1C-F). When we count absolute number of cells from the different progenitor populations per individual yolk sac we observe no differences in CD41+ckit+ hematopoietic progenitors. Therefore, we can now state that *Nanog* expression in the embryo does not lead to an overall increase in hematopoietic progenitors. We have corrected the text to this end, and replaced the original relative data in Fig 1 with absolute cell number per yolk sac.

As for the increased expression of *Runx1* we observe in the E9.5 embryo, we believe it could be a stage-specific event related to the time when we express *Nanog*, as we do not observe this effect in either E7.5 embryos (Fig EV1F) or in the adult MEPs (Fig 4F).

- Tal1 expression is initially assessed in the small subpopulation of CD71+ter119+ cells that can develop, and the decrease appears modest. One could assume that more drastic downregulation could potentially be observed at earlier Flk1 mesoderm. This appears to be the case based on Fig 5 where more complete downregulation of Tal1 is observed when the early embryos were analyzed as a whole. However, Tal1 is not only erythroid cell specifier, but its complete loss leads to loss of CD41+ckit+ cells in the yolk sac and embryo. If Tal1 expression in mesoderm is repressed by *Nanog*, why is there increase in CD41+ckit+ cells? Can the authors comment on these discrepancies?

As mentioned above, the increase we described for this population was only in relative numbers as we were showing percentage of total cells in the original version of the manuscript. We are now showing absolute cell number per yolk sac, and observe no differences in CD41+ckit+ cells (revised Fig 1F), and corrected the text accordingly.

- A comment "Gain of *Nanog* expression in mature erythrocytes (CD71+Ter119+ population) resulted in significant downregulation of Tal1 and upregulation Runx1" is not completely correct as they still retain CD71 expression and nucleus. Mature erythrocytes, even primitive, are identified by enucleation.

This is clearly a mistake we made in the text, as we generally refer to this population as erythroblasts. We thank the referee for pointing it out and have corrected this error.

- It would be helpful to show vascular staining also in the Yolk sac to provide evidence that vascular development in hematopoietic sites is normal as their regulation in hemogenic vasculature could be different.

We appreciate this nice suggestion of the referee, and have carried out CD31 immunohistochemistry on the yolk sac of untreated and dox-treated E9.5 embryos, observing proper vasculature development in yolk sacs. We have added this data to Fig EV1A and discussed it in the text.

- As a general phenomenon, the figures are not very informative alone as they lack information about tissues used for the analysis, embryonic age etc. Adding some detail would make it easier to follow both for reviewers and readers as figure legends are not always available in the same page.

We agree with the referee on this point and have labeled figures accordingly to make their comprehension easier to follow.

- Does the heart beats normally? Given that heart develops from Flk1 mesoderm similar to blood and vascular lineages, and proposed *Nanog* target *Tal1* is known to repress cardiac fate, it would be good to give a general idea whether the heart development is generally unaffected.

We have not observed defects in heart development in these embryos. Hearts beat normally in freshly dissected embryos and the overall morphology of the heart is normal. Furthermore, we do not observe changes in the expression of Flk-1 (*Kdr*) in E7.5 embryos (Fig. EV1F). We have included in Fig EV1B H&E stained sections of E9.5 embryos to show in more detail how cardiac development in *Nanog* expressing embryos is normal (see also E10,5 embryos shown above in Fig R1A), and also included a comment in the main text.

- The reduction in *Tal1* levels in the BM erythroblasts does not appear very drastic and unlikely to explain the defect as a whole, as *Tal1* heterozygous mice do not have obvious erythroid defect. The accumulation of CD71+Ter119+ erythroblast and minimal evidence of enucleated erythroid cells is a strong phenotype, but it is unclear how this is caused. Genomewide gene expression profiling would likely provide a more comprehensive picture and might identify programs that are not even considered otherwise.

As mentioned above, we have carried out RNA-seq on the MEP population to have a broader sense of *Nanog* effects on the transcriptome. We observe that genes downregulated in MEPs from dox-treated animals are enriched in functional terms related to bone marrow cell populations, and more specifically MEPs. This confirms that *Nanog* is repressing the transcriptional program for erythroid progenitors. On the other hand, genes that are upregulated upon *Nanog* induction are highly enriched in the mast cell program. This is very interesting, as it has been shown that deletion of *Tal1* during adult hematopoiesis results in production of mast cells from MEPs, while under normal conditions these cells derive from granulocyte-monocyte progenitors (Salmon et al. Blood 2007). This is accompanied by an upregulation of *Gata2*, gene that we see increased upon *Nanog* overexpression in MEPs. Furthermore, the

expression of *Cebpa*, a factor that represses mast cell lineage (Iwasaki et al. Genes Dev 2006) is downregulated in the *Nanog*-expressing MEP population.

Therefore, the transcriptomic analysis of MEPs confirms that the down-regulation of the erythroid program is a prime target of *Nanog*, but also has provided novel insight into the phenotype that we had not identified previously. However, we believe that the positive regulation of the mast cell program is not a physiological role of *Nanog*, because this cell type does not appear during gastrulation (as erythroid progenitors do) but later in the yolk sac and the AGM (Gentek et al. Immunity 2018) where *Nanog* is not expressed. Thus, we consider that up regulation of the mast cell program is a secondary consequence of the downregulation of erythroid lineage factors, such as *Tal1*, in *Nanog* expressing MEPs. We have included these results in the revised version of the manuscript and presented the data in a new supplementary figure (Fig EV3).

As for the changes in different populations, we have opted to show absolute cell number per femur instead of relative percentages, following the above suggestion of the referee. We observe a similar result, with a decrease of LSK progenitors and CMPs, together with a strong increase in MEPs (revised Fig 4E). To further characterize the adult phenotype, we have also carried out transplants of *Nanog*^{tg} bone marrow into wild type recipients, and observed that after 4 months of dox treatment, *Nanog*-expressing MEPs were completely outcompeted by wild type cells (revised Fig 4G-I). Together, these results reinforce the idea that *Nanog* is acting specifically on MEPs, by blocking their differentiation what leads to their accumulation. These progenitors are not capable of completing their differentiation and are therefore replaced by wild type cells in a competitive setting. These new results have been included in the revised version of the manuscript and discussed accordingly.

- EV.1 Increase of *Tal1* in *ckit*-CD41+ cells in Dox treated embryos is difficult to interpret as it is not clear what cells are included in this population. But this results generally would argue against *Tal1* being a main direct target that is repressed by *Nanog*.

Although there is an apparent increase of *Tal1* in this cell population, that includes both erythroblast and megakaryocyte precursors, the difference between untreated and dox treated embryos is not statistically significant.

- EV4 gel needs labeling

We are sorry for the poor labeling of the gel, that we have now labeled properly indicating the sizes of the wild type and deleted PCR bands (revised Fig EV5B).

2nd Editorial Decision

17th Jan 2019

Thank you for submitting the revised version of your manuscript. My apologies again for the unusual delay with the reassessment of your work. Your revised study has now been re-evaluated by two of the three original referees, please find their comments enclosed below. As you will see the referees find that their concerns have been sufficiently addressed and they are now broadly favourable of publication.

Thus, we are pleased to inform you that your manuscript has been accepted in principle for publication in The EMBO Journal, pending some minor issues regarding literature cited, formatting and data representation, as outlined below, which need to be adjusted at re-submission.

REFEREE COMMENTS:

Referee #2:

The authors have substantially added to the paper, in particular increasing sample sizes on key experiments, and performing replicates in independent cell lines. The data are now in my opinion quite solid that Nanog is a repressor of SCL. One minor point: In my reading, the initial study showing that CD41 marks developmental hematopoietic progenitors was:

Mitjavila-Garcia, M. T. et al. Expression of CD41 on hematopoietic progenitors derived from embryonic hematopoietic cells. *Development* 129, 2003-2013 (2002).

Referee #3:

The authors have done extensive new experiments to respond to the reviewer comments. I am satisfied with the response to my questions and concerns.

2nd Revision - authors' response

24th Jan 2019

All requested editorial changes were made.

Corresponding Author Name: Miguel Manzanares

Manuscript Number: EMBOJ-2018-99122